

**Can the combining of wetlands with reservoir operation largely reduce the risk of**
**future flood and droughts?**
Yanfeng Wu[1], Jingxuan Sun[1,2], Boting Hu[1,2], Y. Jun Xu[3], Alain N. Rousseau[4], Guangxin Zhang[1,*]
[1] Northeast Institute of Geography and Agroecology, Chinese Academy of Sciences, Changchun, Jilin
130102, China
[2] University of Chinese Academy of Sciences, Beijing 100049, China
[3] School of Renewable Natural Resources, Louisiana State University Agricultural Center, 227 Highland
Road, Baton Rouge, LA 70803, USA
[4] INRS-ETE / Institut National de la Recherche Scientifique - Eau Terre Environnement, 490 rue de la
Couronne, G1K 9A9 Quebec City, Quebec, Canada
* Correspondence: Professor Guangxin Zhang (zhgx@iga.ac.cn)
**Abstract.** Wetlands and reservoirs are important water flow and storage regulators in a river basin;
therefore, they can play a crucial role in mitigating flood and hydrological drought risks. Despite the
advancement of river basin theory and modeling, our knowledge is still limited about the extent that
these two regulators could have in performing such a role, especially under future climate extremes. To
improve our understanding, we first developed a framework coupling wetlands and reservoir operations
with a semi-spatially explicit hydrological model and then applied it in a case study involving a large
river basin in Northeast China. The projection of future floods and hydrological droughts was performed
using this framework during different periods (near-future: 2026-2050, mid-century: 2051-2075, and
end-century: 2076-2100) under five future climate change scenarios. We found that the risk of future
floods and hydrological droughts can vary across different periods, in particular, will experience
relatively large increases and slight decreases. This large river basin will experience longer duration,
larger peak flows and volume, and enhanced flashiness flood events than the historical period.
Simultaneously, the hydrological droughts will be much more frequent with longer duration and more
serious deficit. Therefore, the risk of floods and droughts will overall increase further under future
climate change even under the combined influence of reservoirs and wetlands. These findings highlight





the hydrological regulation function of wetlands and reservoirs and attest that the combining of wetlands
with reservoir operation cannot fully eliminate the increasing future flood and drought risks. To improve
a river basin's resilience to the risks under future climate change, we argue that implementation of
wetland restoration and development of accurate forecasting systems for effective reservoir operation
are of great importance. Furthermore, this study demonstrated a wetland-reservoir integrated modeling
and assessment framework that is conducive to risk assessment of floods and hydrological droughts,
which can be used for other river basins in the world.
**Keywords**: Climate change; Hydrologic projection; Floods and droughts; Wetland hydrological
services; Reservoir operations; Model integration
**1. Introduction**

Floods and droughts have produced some of the most frequent and serious disasters in the world

(Hirabayashi et al., 2013; Unisdr, 2015; Diffenbaugh et al., 2015a). Globally, they account for 38% of
the total number of natural disasters, 45% of the total casualties, more than 84% of the total number of
people affected, and 30% of the total economic damage caused by all-natural disasters (Güneralp et al.,
2015) in the past. In the future, as climate change has been accelerating the hydrological cycle, causing
more frequent and stronger weather extremes, more floods and droughts have been projected to increase
at both regional (Hallegatte et al., 2013; Wang et al., 2021) and global scales (Jongman, 2018; Chiang
et al., 2021). Concurrently, the loss of disaster-related ecosystems (e.g., wetlands, forest and grassland)
and their services can cascade up the flood and drought risks to a great extent (Gulbin et al., 2019; Walz
et al., 2021). Given this, grey infrastructure such as dams, dikes, and reservoirs, which have often been
used to attenuate flood and drought hazards because of their rapid and visible effects, can play an
important role in ensuring the water security of a river basin (Casal-Campos et al., 2015; Alves et al.,
2019). However, relying solely on grey infrastructure to attenuate floods and droughts has some
inadequacies, such as large investments to build and maintain in addition to adverse effects on
downstream ecosystems (Maes et al., 2015; Schneider et al., 2017). In this context, Nature-based



solutions (NBS) for hydro-meteorological hazards mitigation are becoming increasingly popular
(Kumar et al., 2021), because NBS can effectively reduce or even offset the hydrological processes
driving floods and droughts (Nika et al., 2020), while making least disturbance to the environment as
well as delivering co-benefits which grey infrastructure cannot provide (Nelson et al., 2020; Anderson
and Renaud, 2021). Therefore, it is urgent to incorporate NBS in the current water management practices
to increase basin resilience to hydrological extremes under future climate change.

Wetlands have the potential to be used as a nature-based solution for improving water storage and

hence the resilience of a river basin to hydrological extremes along with grey infrastructures (Thorslund
et al., 2017). This is because, similar to man-made dams and reservoirs, wetlands can attenuate flow and
alter basin hydrological processes (Lee et al., 2018), such as floods (Wu et al., 2020a) and baseflows
(Evenson et al., 2015; Wu et al., 2020b). Unlike man-made grey infrastructures, wetlands are integral
in landscapes and they are connected laterally and vertically with the surrounding terrestrial and aquatic
environments through the hydrological cycling of water and waterborne substances (Ahlén et al., 2020),
making their water storage and cycling fundamental to estimate a watershed's water balance (Golden et
al., 2021; Shook et al., 2021). To understand how and to what extent wetlands can mitigate basin
hydrological processes, several wetland hydrological models have been developed and applied to
quantify hydrological functions of wetlands, particularly the mitigation services on floods and droughts.
For instance, Ahmed(2014) modified three parameters of NAM module in Mike11 model, i.e. the
maximum water content in surface storage, maximum water content in root zone storage and overland
flow runoff coefficient, to discern the cumulative effect of wetland loss on flood peak flow and low
flows. Wang et al.(2008) incorporated wetlands into SWAT model using a hydrologic equivalent
wetland (HEW) concept and Liu et al.(2008) developed an extension module for delineating riparian
wetland hydrology and embedded it into SWAT model. Since then, Evenson et al.(2016), Evenson et
al.(2018), Lee et al.(2018), Chen et al.(2020), Zeng et al.(2020) successively modified wetland modules
(isolated or riparian wetlands) and improved the applicability of SWAT model to discern hydrological
services of basin wetlands. Fossey et al.(2015) integrated two wetland modules (isolated and riparian
wetlands) into the PHYSITEL/HYDROTEL modelling platform and then investigated the impacts of



wetland geographic location and typology on high and low flows (Fossey et al., 2016). These wetland
hydrological models not only consider the general water budget of a river basin but also take into
account the perennial and intermittent hydrological interactions between wetlands-to-wetlands and
wetlands-to- surrounding landscapes. It is of both scientific and practical interest to assess the models
and insights arrived from them for projecting wetland capability in mitigating floods and droughts in
response to a changing climate.

Reservoirs redistribute large amounts of surface water, thus altering natural hydrological processes,

such as flow range, flood and drought patterns, and basin water balances (Zhao et al., 2016; Boulange
et al., 2021; Chen et al., 2021; Manfreda et al., 2021). So far, throughout the world, there are 57, 985
reservoirs registered by the International Commission on Large Dams and their total volume has been
reached 14, 602 km$^3$ (Eriyagama et al., 2020). Such numerous reservoirs and their large storage capacity
should not be neglected in water hazard assessment and hydrological projection because of their
significant modification on river flow regimes. A recent study by Brunner(2021) evidenced that
reservoir regulation can modulate flood and drought patterns by reducing drought severity and duration,
as well as altering spatial flood connectedness. Boulange et al.(2021) pointed out that consideration of
reservoirs can significantly affect the estimation of future population exposure to flood. They called for
the need to integrate reservoirs in model-based impact analysis of flood exposure under climate change.
Dang et al. (2020a) and Yassin et al. (2019) therefore argued that failure to represent these effects could
limit the performance of hydrological models and suppress the applicability of such models to support
basin flood and drought mitigation practices. Considering the increasing number of reservoirs and the
requirement for more accurate management practices, there is a growing need in incorporating reservoir
operations into basin hydrologic simulations and predictions.

Despite the well-established knowledge of flow and storage regulation functions that wetlands and

reservoirs can provide in a river basin, most modeling assessments on floods and droughts at the basin
scale do not take the two components into account, or give little emphasis on the combined benefits of
them (Brunner et al., 2021; Golden et al., 2021). Nor are the hydrological processes associated with
these features implicitly including in the calibration of hydrologic models. Recent studies have

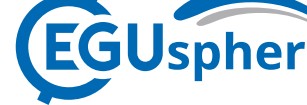

suggested that doing so would add significant error and larger uncertainties to simulate hydrologic
processes (Ward et al., 2020; Brunner et al., 2021); while integrating the wetlands or reservoir operation
alone into watershed-scale hydrologic models may largely minimize uncertainties (Zhao et al., 2016;
Dang et al., 2020; Rajib et al., 2020a; Golden et al., 2021) and improve model performance (Liu et al.,
2008; Fossey et al., 2015; Evenson et al., 2016; Yassin et al., 2019). Furthermore, on a global scale,
most river basins have wetlands and their river flow has or will experience reservoir regulation
(Schneider et al., 2017; Muller, 2019), which elicits two thought-provoking concerns: Can coupling
wetlands and reservoirs for hydrological modeling achieve a '1+1=2' simulation effect to support policy
decisions? If yes, what will be the changes of future floods and droughts under the combined influence
of wetlands and reservoirs? Such concerns are important because the omission of wetlands and
reservoirs can cause the policy-making process to be imprecise at best and ineffective at worst. However,
a reservoir operation and wetland services, integrated basin-scale model rarely exist in the literature, nor
is it clear how floods and droughts will be changed under future climate change.

In a very recent study conducted in the Nenjiang River Basin, Wu et al. (2022) quantified wetland

flood mitigation services under future climate change and reported that future precipitation extremes
will cause flood risks that cannot be mitigated by wetlands. They calibrated hydrological model coupling
isolated wetlands and riparian wetlands but didn't consider damming effects on flood risks and wetland
hydrological function. However, reservoir operation has largely altered downstream flooding processes
(Chen et al., 2021) and reduced flow regulation service of downstream wetlands to some extent in the
river basin (Wu et al., 2021). For another recent study, Rajib et al. (2020) incorporated surface
depression and wetland water storage into hydrologic model in the upper Mississippi River Basin and
found that depression-integrated model improved streamflow simulation accuracy with increasing
upstream abundance of depression storage. They also parameterized 15 major lakes and reservoirs using
their unique hydraulic design and storage-discharge information. These two studies provide insights into
modeling and understanding the flow and storage regulation functions provided by wetlands and
reservoirs, however, is it still unclear whether the combining of wetlands with reservoir operation can
largely reduce the risk of future floods and droughts.



Considering the above-introduced scientific challenges and management deficiencies, we first
developed a framework of hydrological modeling coupled with wetland modules and reservoir operation
scenarios. We applied it to a large river basin with abundant wetlands and a large reservoir, the Nenjiang
River Basin in northeast China, to address a central question: How will estimated future flood and
drought risks be changed by considering the combined effects of wetlands and reservoirs? We addressed
these questions by (a) assessing performance of hydrological modeling framework from the perspective
of streamflow processes and hydrography as well as flood and drought characteristics, (b) projecting
flood and drought risks in terms of their characteristic indicators using the hydrological modeling
framework, and (c) discussing our findings and implication for practical flood and drought risk
management. Our framework and results are expected to bring new insights into future floods and
droughts and provide a basis for decision-making to curb the growing impacts of unprecedented and
future extreme conditions.
**2. Methodology**
2.1 Study area
We conducted this analysis in the Nenjiang River Basin (NRB), a large river basin (291,700 km$^2$)
located in the Northeast China (Fig. 1). Long-term annual average runoff depth and volume from the
NRB are 97.4 mm and 22.7 billion m$^3$. The river basin is located in the middle-high latitudes and can
be characterized by a temperate semi-humid continental monsoon climate. Inter-annual differences in
temperature and precipitation are large, i.e. disparate hot and cold periods, and uneven dry and wet
conditions (Meng et al., 2019). The average annual temperature across the basin ranges between 2.1-
4.5°C. The annual total precipitation within the basin fluctuates from 323.1 to 537.6 mm. Precipitation
is mainly concentrated during June-September, which accounts for about 85% of the annual
precipitation (Li et al., 2014).
The NRB is one of the pivotal wetland areas in China. Many wetlands in the river basin have been
designated as a Ramsar Site of International Importance, including the Zalong, Xianghai, Momog and
Nanwen wetlands. The wetlands and their contributing drainage areas within the reaches monitored by





the ten hydrological stations range from 14 to 23% and from 39 to 56% respectively, demonstrating the
large wetland coverage of the NRB and its sub-basins (Table 1). The lower NRB is an important
agricultural area of the Songnun Plain, which is one of the three major plains (including the Sanjiang,
Songnun and Liaohe Plains) in northeast China. The Songnum Plain is also a crucial food production
base in China. Therefore, understanding potential floods and hydrological droughts under future climate
change is crucial for ensuring regional food security and ecological integrity. During the past 60 years,
land use and land cover types have drastically changed owing to large-scale development of intensive
agriculture and water resources management (Meng et al., 2019). The area of wetlands has significantly
decreased and their services have been degraded. For example, the area of wetlands in the NRB
decreased by nearly 23% from 1978 to 2000 (Chen et al., 2021), with only 16.34% remaining today
(Table 1). Along with the reduction in wetland area, the hydrological functions of wetlands in the NRB,
such as water storage, flood mitigation and baseflow support, have been considerably reduced (Wu et
al., 2021). These wetland services are closely related to flood and drought risks, such as the 1998 mega-
flood. In order to effectively deal with the risk of floods and droughts, the Nierji Reservoir was
constructed along the mainstream NRB (Fig. 1); it started normal operation in 2006. The Nierji
Reservoir is located in the upper Nenjiang River (Fig. 1). The reservoir receives inflow from an area of
66,382 km$^2$ and has flood control and water supply as the primary purposes and hydropower generation
and navigation as secondary purposes, thus playing an important role in the distribution of water
resources for the lower NRB.



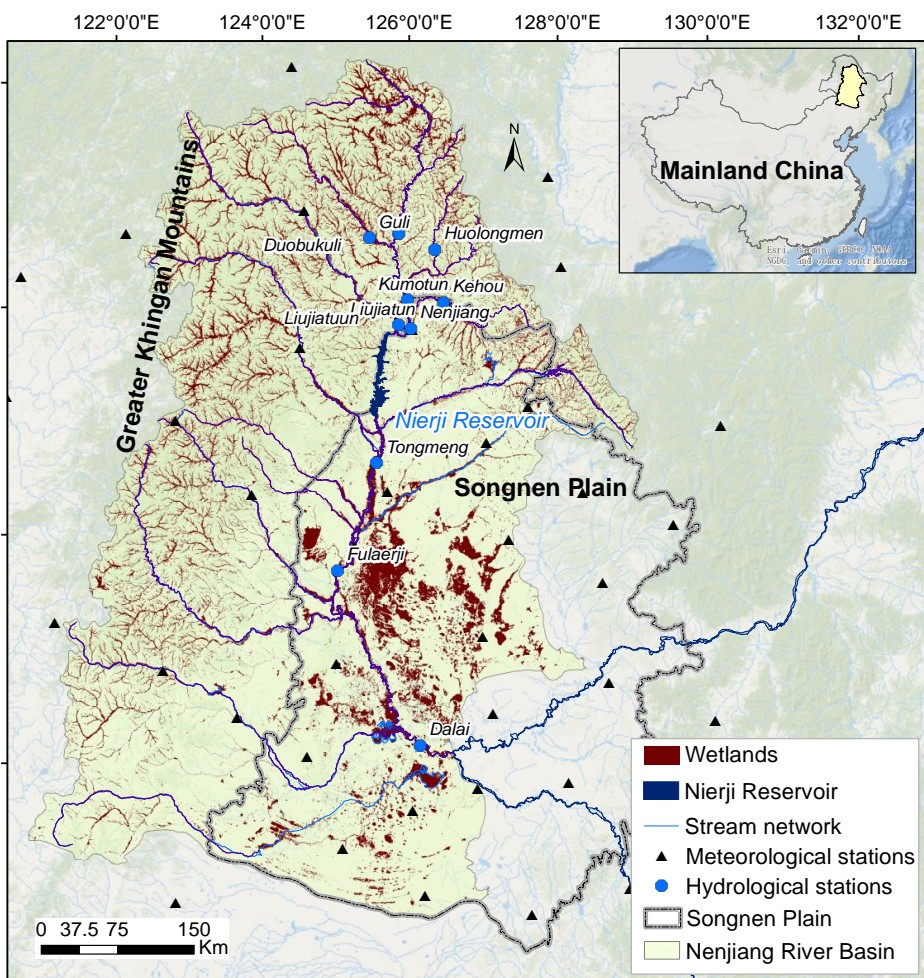

Figure 1. Characteristics of the Nenjiang River Basin in northeast China: (a) location of stream network, Nierji Reservoir, sub-basins, and hydrological and meteorological stations; elevation is also provided; (b) spatial distribution of isolated (IWs) and riparian (RWs) wetlands and their drainage area and (c) land-use types.



Table 1 The drainage area of the ten hydrological stations used in this study, area ratios of wetlands and
their contributing areas to the drainage area of the Nenjiang River Basin, northeast China.

| ID | River | Hydrological stations | Drainage area (km$^2$) | Wetland area ratio (%) | Wetland contribution area ratio (%) |
|----|-------|-----------------------|------------------------|------------------------|-------------------------------------|
| 1  | Mainstream | Shihuiyao | 17205 | 22.2 | 54.7 |
| 2  | Duobukuli River | Guli | 5490 | 16.3 | 57.1 |
| 3  | Menlu River | Huolongmen | 2151 | 20.8 | 50.7 |
| 4  | Mainstream | Kumotun | 32229 | 20.4 | 54.3 |
| 5  | Keluo River | Kehou | 7310 | 23.4 | 56.2 |
| 6  | Gan River | Liujiatun | 19665 | 13.2 | 49.9 |
| 7  | Mainstream | Nenjiang | 61249 | 18.3 | 54.1 |
| 8  | Mainstream | Tongmeng | 108029 | 13.1 | 47.5 |
| 9  | Mainstream | Fulaerji | 123911 | 13.7 | 39.0 |
| 10 | Mainstream | Dalai | 221715 | 16.3 | 42.4 |


2.2. Overview study approach
The methodological framework proposed in this paper includes two parts (Fig.2): (a) coupling
wetlands and reservoir operations with basin hydrological processes simulation, and (b) projection of
future flood and drought characteristics under different climate scenarios. Specifically, we first
developed a semi-spatially explicit hydrological model that considers wetland hydrological processes
and reservoir operations through coupling a distributed hydrological modeling platform with wetland
modules and reservoir simulation algorithms (see Part I in Fig.2 and Sect. 2.3). Then, the distributed
hydrological modeling platform was used to simulate streamflow driven by multi-model ensemble
means from the latest CMIP6 and to derive drought and flood characteristics (see Part II in Fig.2 and
Sect. 2.4). The flood and drought characteristics were then compared against historical periods to discern
how future hydrological extremes will be changed under the influence of wetlands and reservoirs.



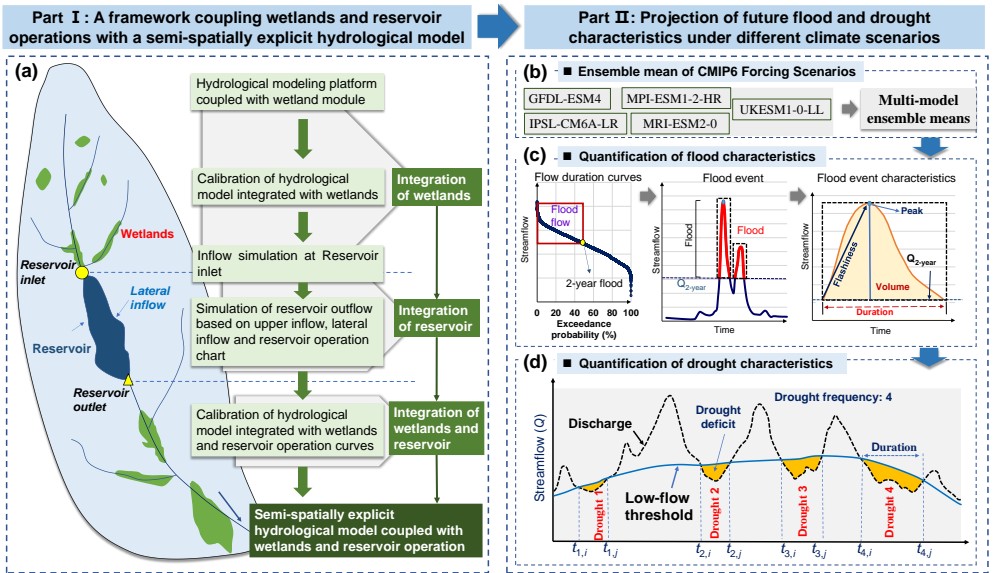

Figure 2. Framework for projecting future flood and hydrological droughts based on a semi-spatially

integrating wetlands and reservoir operation into a hydrological model: (a) a framework coupling

wetlands and reservoir operations with a semi-spatially explicit hydrological model; (b) multi-model

ensemble means from five GCM projections used for driving modeling framework; (c) methodology

for determining a flood threshold, defining flood events, and extracting flood characteristics, and (d) a

sequence of runs with examples of drought deficit, duration, and frequency.

## 2.3. Framework of hydrological modeling coupled with wetland modules and reservoir operation scenarios

We calibrated the model with measurements collected upstream of the reservoir inlet. The calibrated,

coupled hydrologic model (i.e. hydrologic-wetland model) was then used to simulate inflow to the upper

reservoir. Simultaneously, we estimated the lateral inflow into the reservoir. Based on the simulated

runoff at the inlet, lateral inflow, and the schemes of reservoir operation, we estimated the reservoir

outflow using the reservoir simulation algorithms. The simulated runoff simulated by hydrologic-

wetland model at the reservoir outlet was replaced with the estimated reservoir outflow, thus integrating

reservoir operation into the hydrological modeling (i.e. hydrologic-wetland-reservoir model). The



replaced runoff was used to calibrate the hydrologic-wetland-reservoir model for the lower reach of the
reservoir, thus integrate the downstream wetlands into the hydrologic model. Based on this framework,
the simulation of basin hydrological processes coupled with wetlands and reservoir operations were
realized.
2.3.1. A semi-distributed hydrological model platform coupled with wetland modules

We used the PHYSITEL/HYDROTEL modeling platform (Fossey et al., 2015), coupled with two

wetland modules, to simulate hydrological processes, assess model performance and project future flood
and drought conditions. This platform had been used to quantitatively evaluate the hydrological function
of wetlands by Fossey and Rousseau (2016), Fossey and Rousseau (2016), Blanchette et al. (2019), Wu
et al.(2020)a , Wu et al.(2020)b, Wu et al.(2021) and Blanchette et al.(2022). PHYSITEL is a
Geographic Information System based pre-processing platform for managing hydrological modeling
data (Rousseau et al., 2011; Noël et al., 2014). Using general basin data (a digital elevation model,
vectorized river network and lacustrine water bodies, and raster-based land use and soil matrix
distribution maps), PHYSITEL divides the basin into more detailed hydrological response units, i.e.
relatively homogeneous hydrological units (RHHUs) (Fortin et al., 2001). The RHHUs were defined
using the algorithm for delineating and extracting hillslopes proposed by Noël et al.(2014). The
hillslopes with same characteristics (e.g., physical geography and hydrological response) were then
aggregated within each RHHUs. In addition, the PHYSITEL platform distinguishes wetlands from other
land-use types, and then classifies both isolated and riparian wetlands based on the percentage of pixels
adjacent to the hydrographic network (Fossey et al., 2015). It subsequently generates data pertaining to
isolated and riparian wetlands and their contributing areas (CA). The PHYSITEL platform uses the
concept of a hydrologically equivalent wetland (HEW) proposed by Wang et al.(2008) to integrate
isolated wetlands (IWs) and riparian wetlands (RWs) at the RHHU scale. These typically large RHHUs
contain large wetland complexes consisting of various wetland categories such as bogs, fens, marshes,
and forested peatlands. After defining the hydrological and wetland parameters, PHYSITEL can directly
export the database as part of the input data to HYDROTEL; these data can also be used for other
watershed hydrological models.



HYDROTEL is a physically-based and semi-distributed hydrological model (Turcotte et al., 2007;
Bouda et al., 2012; Bouda et al., 2014) that requires wetland parameter data, land-use type maps, soil
texture maps, meteorological data (e.g., daily temperature and precipitation) and daily flows as input.
The HYDROTEL model couples the hydrological processes associated with both IWs and RWs (i.e. the
IWs and RWs modules) at the RHHU scale and calculates the wetland water balance with respect to the
surface area of the HEW, CA and RHHU. Specifically, for IWs, the hydrogeological processes are
integrated in the vertical water budget (Fortin et al., 2001) at the RHHU scale. For RWs, the water
balance is partially integrated in the vertical water budget of an RHHU and directly connected to the
associated river segment via the kinematic wave equation. Based on this, the IWs modules can realize
the vertical water balance processes of hillslope wetlands with land surface runoff processes, while the
RWs modules can realize the interaction of hydrological processes between RWs and river channels.
These representations provide a modelling approach that can simulate water balances at the wetland
scale while considering their interactions with the surrounding environment (contributing drainage area
and hydrological connectivity) (Fossey et al., 2015).
2.3.2. Simulation of Nierji reservoir operations
We used the designed operating curves of the reservoir operation chart and the ResSimOpt-Matlab
software package developed by Dobson et al.(2019) to simulate the operation of the Nierji Reservoir.
ResSimOpt-Matlab contains three algorithms for reservoir simulation. A dynamic operation schemes
was used in this study to achieve the simulation. Specifically, following Dobson et al.(2019) and
according to actual hydrological conditions, we defined two seasons: the wet season (from June to
September) when the risk of flooding is higher and we wanted to release the target demand and provide
some storage space for flood control, and the dry season when the risk of flooding is low and the main
objective is to sustain ecological baseflows. The required input data to the algorithm includes reservoir
inflow ($Q_{in}$), the minimum environmental flow ($E_{env}$), initial storage ($S_{o}$), minimum ($S_{min}$) and maximum
($S_{max}$) storage, estimated evaporative losses ($E_{vap}$), released discharge ($Q_{out}$) and the simulation time-
step length. Based on the required data, we performed reservoir simulation by implementing the mass
balance equation at each simulation time step $t$:



$$\begin{cases} S_{(t+1)} = S_{(t)} + Q_{in(t)} - E_{vap(t)} - Q_{out(t)} \quad or \quad S_{(t)} + Q_{in(t)} - E_{\min(t)} - E_{vap(t)} \\ 0 \le S_{(t)} \le S_{\max} \\ 0 \le R_{(t)} \le \min\left(S_{(t)} + Q_{in(t)} - E_{\min(t)} - E_{vap(t)}, Q_{\max}\right) \end{cases} \quad (1)$$

where $S_t$ is the reservoir storage at time $t$. $S_t$ and $Q_{out}$ are constrained by the design specifications
and operation rules of a reservoir. Specifically, $S_t$ cannot exceed the reservoir capacity $S_{max}$, while
$Q_{out}$ is constrained by the operation schemes and capacity of the turbines $Q_{max}$. The excess water, if
any, is spilled:
$$Q_{spill(t)} = \max\left(S_{(t)} + Q_{in(t)} - E_{vap(t)} - Q_{out(t)}\right) \quad (2)$$

Based on this, the dynamic $Q_{out}$ can be represented using the equation (1) and (2).
We collected information on the reservoir operation including reservoir capacity, control water levels,
outflow, the storage-area-water level relationship, the tailwater level-discharge relationship, and the
maximum release, along with other data necessary to estimate the outflow. The reservoir inflow is the
simulated streamflow at the Nengjiang Hydrological Station, which is at the inlet of the Nierji Reservoir.
The minimum storage and maximum storage are 4.9 billion $m^3$ and 86.1 billion $m^3$, respectively. Based
on the available data for the study area, the Karrufa method (Kharrufa, 1985) was used to estimate daily
evaporative losses from the reservoir. We convert days to seconds so that it would correspond to the
flow data. During the wet season, the actual operation schemes for the Nierji Reservoir are as follows:
June 1-20 is the pre-flood period with a flood limited water level of 216.0 m; June 21-August 25 is the
main flood period and the reasonable flood limited water level ranges from 213.4 m to 216.0 m and can
be gradually increased. September 6-30 is the post-flood period with a flood limited water level of 216.0
m. During the dry season, the environmental flow was defined as 25.3% of the daily streamflow based
on the designed operating curves of the reservoir operation chart.
2.3.3 Driving datasets, model calibration and performance assessment
The driving datasets used in this study include meteorological data, land-use/land-cover types, soil
texture, digital elevation models, drainage network, and observed discharge data. The land-use/land-
cover types for 2015, soil texture, digital elevation models, digital elevation models and drainage



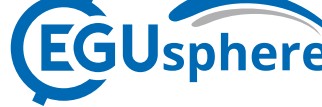

network were obtained from Resource and Environment Science and Data Center
(https://www.resdc.cn/). We collected the wetland distribution maps for 2015 (extracted from National
wetland map in China produced by Mao et al., 2020) and overlaid it with the land-use/land-cover types.
Historical daily meteorological datasets including precipitation and air temperature for the period 1963-
2020 were obtained from 39 weather stations administered by the National Meteorological Information
Centre of China (http://data.cma.cn) and 49 weather stations in the upper NRB (Fig. 1) administered by
the Nenjiang Nierji Hydraulic and Hydropower Ltd. Company (http://www.cnnej.cn). The hydrological
data from ten hydrological stations were obtained from the Songliao Water Resources Commission,
Ministry of Water Resources (http://www.slwr.gov.cn/), with the time series extending from 1963 to
2020. Of the ten stations, seven are located upstream of the Nierji Reservoir.

We used observed streamflow of ten hydrological stations to calibrate the HYDROTEL model. Seven

hydrological stations (Shihuiyao, Guli, Huolengmen, Kumotun, Kehou, Liujiatun and Kumotun) are
located upstream of the Nierji Reservoir, and the rest stations (Tongmeng, Fulaerji and Dalai) are
installed at downstream of the reservoir. For the upstream Nierji Reservoir, we calibrated the
HYDROTEL model against observed streamflow of seven hydrological stations under with and without
wetland scenarios. Among the seven hydrological stations, the Nenjiang Station is located at the end of
the upstream, where the simulated streamflow was taken as the inflow of the reservoir. For the
downstream Nenjiang Station, we calibrated HYDROTEL model in the presence or absence of the
combination of wetlands and reservoir, respectively. In the case of the combination of wetland and
reservoir, we first simulated the operation of the Nierji Reservoir and calculated the outflow of the
reservoir (Sect. 2.3.2), which was used as the input streamflow for downstream model calibration. We
then calibrated HYDROTEL model against observed streamflow of Fulaerji and Dalai Stations under
with and without wetlands scenarios, respectively. Note that the without wetland scenarios are defined
as follows: When the wetland modules are turned off in HYDROTEL, wetland areas are not removed,
but they are treated as the land cover of saturated soils and thus their explicit storage properties and
hydrological dynamics are not accounted for in the modeling (Wu et al., 2020a). This is a basic
assumption that has been used in several studies using models such as SWAT (Liu et al., 2008; Wang et





al., 2008; Evenson et al., 2015), Mike 11 (Ahmed, 2014) and HYDROTEL (Fossey et al., 2016; Fossey
and Rousseau, 2016a, b; Wu et al., 2019, 2020a, 2021), to quantify the hydrologic services provided by
wetlands (flood mitigation, flow regulation and baseflow support etc.).

For all above scenarios, we calibrated the HYDROTEL model against observed streamflow at a daily

time step over 8 years, including a 1-year warm-up (2010.10.01-2011.09.30) and a 7-year calibration
(2011.10.01-2018.09.30) periods. The same model settings (i.e. key parameters, simulation periods,
fitting algorithm, and objective function, etc.) were used for the calibration processes under the both
presence and absence scenarios. Following Arsenault et al.(2018), the model was calibrated using full-
time observations without additional validation, as the former allows for more reliable parameters and
maximizes the accuracy of the model. The dynamically dimensioned search algorithm (DDS) developed
by Tolson and Shoemaker(2007) was used to calibrate the 13 most sensitive parameters of the model as
proposed by Foulon et al.(2018). Based on the maximizing of Kling-Gupta efficiency (KGE) (Gupta et
al., 2009), automatic calibrations using DDS were carried out utilizing 10 optimization trials. (250 sets
of parameters per trial). Then, the best set of parameter values out the 10 trials were selected. The KGE
was chosen as the objective function because previous research has shown that it can improve flow
variability estimates when compared to the NSE (Garcia et al., 2017; Fowler et al., 2018).

To determine whether coupling the wetland module and the reservoir can improve the model

performance, we compared (1) the efficiency of the model in simulating daily flow processes; and (2)
the capability of the model to simulate floods and hydrological droughts in the presence or absence of
the wetlands and the combination of wetlands and reservoir. Following the recommendations of N.
Moriasi et al.(2007) and Moriasi et al.(2015), four objective functions were selected to assess model
performance with regards to simulated daily flows with and without the presence of the wetland modules
and reservoir operation, namely the Nash-Sutcliffe efficiency (NSE) (Nash and Sutcliffe, 1970),
Correlation Coefficient (CC), the root-mean square error (RMSE) and the percent bias (Pbias). We used
multiple objective functions because it may unreliable to rely on a single objective function to determine
whether the model performs well (Pool et al., 2018; Fowler et al., 2018; Seibert et al., 2018). In addition,
we compared model performance considering daily hydrograph changes. Furthermore, flood and





drought features were extracted (see Sect. 2.4.2 and 2.4.3) and used to discern whether, and to what
extent, the coupled wetland modules and reservoir simulations could improve the model's ability to
simulate droughts and floods.
2.4. Projection of future flood and drought characteristics under different climate scenarios
The future simulated streamflow at the Nenjiang and Dalai hydrological stations driven by the
ensemble mean of bias-corrected CMIP6 Forcing Scenarios were selected to derive drought (i.e. number
of droughts, annual drought days, duration and deficit of each drought) and flood (i.e. peak, duration,
volume and flashiness) characteristics. The Nenjiang Station was chosen because it is located at the
outlet to (mouth of) the upper NRB and the inlet to the Nierji Reservoir, whose flood and drought
patterns are mainly driven by wetlands and climate change. Moreover, changes in drought and flood
characteristics of the Nenjiang Station are critical to the operation of the reservoir immediately lower
reach. The Dalai Station, located at the outlet of the entire NRB, was used as a proxy to characterize
future flood and drought evolution for the whole basin under the combined influence of the wetlands
and reservoir. Using the calibrated hydrological model that was coupled with wetlands and reservoir
operation, we carried out the simulation of hydrological processes for the historical period (1971-2020)
and under the constraints of the SSP126, SSP370 and SSP585 scenarios. We then extracted flood and
hydrological drought characteristic indices (see sect. 2.4.2 and 2.4.3) from the simulations to conduct a
comparative analysis of their temporal evolution for the near-future (2026-2050), mid-century (2051-
2075) and end-century (2076-2100).
2.4.1 Future climate change scenarios for driving hydrological modeling framework
In this study, we simulated potential future floods and hydrological droughts using five GCM
projections under three Socioeconomic Pathways (SSPs) from the latest CMIP6 (O'Neill et al., 2016).
Each of these specific SSPs represents a development model that includes a corresponding combination
of development characteristics and influences such as population growth, economic development,
technological progress, environmental conditions, equity principles, government management, and
globalization, among others; each also includes a specific description of the extent, speed and direction
of social development. The three SSPs that were used herein include SSP126, SSP370 and SSP585,



which represent potential futures characterized by green-fueled growth (van Vuuren et al., 2017), high
inequality between the countries (O'Neill et al., 2016) and fossil-fueled growth (Kriegler et al., 2017),
respectively. SSP126 is a low forcing scenario with a stable radiative forcing of approximately 2.6 W/m$^2$
in 2100 (van Vuuren et al., 2017). SSP370 is a medium to high radiative forcing scenario with a stable
radiative forcing of approximately 7.0 W/m$^2$ in 2100 (Fujimori et al., 2017). SSP585 belongs to the high
forcing scenario and is the only pathway that achieves emissions as high as 8.5 W/m$^2$ by 2100 (van
Vuuren et al., 2017). These five GCM projections with a high resolution (0.25°) and wide application
(GFDL-ESM4, IPSL-CM6A-LR, MPI-ESM1-2-HR, MRI-ESM2-0, UKESM1-0-LL) were chosen to
provide essential output from the SSPs (Fig.2b).

Given the data requirements of the hydrological model, we downloaded the SSP outputs including

daily precipitation, maximum and minimum temperature. We then performed bias correction and spatial
downscaling of the SSP outputs. The bias correction of SSP outputs was carried out using the CMhyd
software (https://swat.tamu.edu/software/cmhyd), in which the widely used Delta Change method in the
CMhyd software was used. Delta Change bias-corrects the projected SSP outputs based on the historical
statistics and thus conserves the linear spatial-, temporal-, and multi-variable dependence structure in
the future climate (Moore et al., 2008; Bosshard et al., 2011; Maraun, 2016; Shafeeque and Luo, 2021).
The ANUSPLIN package developed by Hutchinson and Xu(2004) was then used to uniformly
downscale the output from five bias-corrected GCMs to a resolution of 1-km based on the DEM.
Following previous studies (Hagemann and Jacob, 2007; Zhao et al., 2021), the multi-model ensemble
means ($M_{GCM}$) of the daily precipitation, and the maximum and minimum temperature under the SSPs
scenarios were then obtained to diminish the uncertainties inherited in a single GCM. MEM was
calculated using an equally-weighted average:
$$M_{GCM} = \frac{1}{N} \sum_{i=1}^{N} P_i \tag{3}$$
where $M_{GCM}$ is the multi-model ensemble means, $N$ is the number of ensemble members (5 in this study);
and $P_i$ is the projected climate data of an ensemble member. In this study, the $M_{GCM}$ of five GCMs were
used to drive hydrological modeling. Future changes in flood and drought characteristics from the



CMIP6 multi-model ensemble mean for the near-future (2026-2050), mid-century (2051-2075), and
end-century (2076-2100) were calculated and compared to the historical period (1970-2018). The
purpose of subdividing the analysis into three time periods was to compare whether, or to what extent,
flood and drought characteristics increase or decrease for different future time periods as compared to a
historical period.
2.4.2 Quantification of flood characteristics

In this study, we characterized floods in terms of four indices consisting of flood peak, flood volume,

duration, and flashiness (Fig. 2c). The 2-year flood streamflow was used as a threshold for defining
flood events, as it has been often used as a substitute of the threshold for bankfull discharge in previous
studies (Cheng et al., 2013; Xu et al., 2019; Wu et al., 2020). Daily streamflows that were greater than
the 2-year flood threshold were considered as flood flows. Flood flows occurring on multiple
consecutive days were considered as a single flood event. The flood indices, i.e., flood peak, volume,
duration, and flashiness were derived with respect to event hydrographs. Flood volume is the cumulative
flow from the initial to the end of a flood event with respect to the 2-year flood streamflow level, and
represents the flood intensity for different flood events (Wang et al., 2015). The annual total flood
volume is the total amount of water associated with all flood events during a water year. We calculated
the annual total flood volume based on flood duration and the average amount of streamflow per event
in a water year. Flood duration varies for different floods and is, therefore, an important characteristic
of a flood event. We summed the flood duration of each event in a water year to obtain the annual flood
days. In addition, the annual maximum peak flow was derived from the daily flows to investigate
changes in the characteristics of extreme floods. We extracted the 2-year flood threshold for a
hydrological station based on the streamflow-exceedance probability curve. Flashiness is a measure of
flood severity and is defined as the difference between the peak discharge and action stage discharge
normalized by the flooding rise time (Saharia et al., 2017).
2.4.3 Quantification of hydrological drought characteristics

We characterized hydrological drought characteristics using four indices consisting of the number of

droughts, annual drought days, drought duration and deficit (Fig. 2d). A threshold method was used to





define hydrological drought events. The threshold method is commonly used because it can
quantitatively determine the start and end of a hydrological drought event, which allows further
assessment of drought characteristics, such as frequency, duration, and intensity of a drought event
(Cammalleri et al., 2017). It is based on defining a flow threshold (discharge, Q, $m^3$/s), below which a
hydrological drought event is considered to occur (also known as a low flow spell). A daily variable
threshold, defined as an exceedance probability of the 365 daily flow duration curves was used to derive
drought events from daily streamflow records (Hisdal and Tallaksen, 2003; Fleig et al., 2006). For rivers
with perennial flow, relatively low streamflows ranging from $Q_{70}$ to $Q_{95}$ have been used as a reasonable
threshold (Zelenhasić and Salvai, 1987; Tallaksen and van Lanen, 2004). In this study, we chose the
90th percentile ($Q_{90}$-n) streamflow as the daily threshold. The $Q_{90}$-n of all days was determined based
on the observed historical daily streamflow. The daily $Q_{90}$-n for each hydrological station obtained in
this way constitutes 365 daily values, excluding the value for February 29th in leap years. The $Q_{90}$-n
values derived from the historical records were also used as the threshold for identifying droughts in
future climate change scenarios.
We analyzed the hydrological drought characteristics based on the hydrological drought threshold
level of $Q_{90}$-n. To enable the comparison across different modeling scenarios (i.e., historical scenarios
and future climate change scenarios), we derived drought days, deficit, duration, and number from
identified hydrological drought events to characterize their patterns. Drought volume deficit was
calculated by subtracting daily streamflow from the threshold level ($Q_{90}$-n) during a drought event, and
it presents the severity of the drought compared to the normal streamflow conditions. Drought duration
was the cumulative number of days during a drought event, i.e., the number of days from the beginning
to the end of the drought. Annual drought days were then the cumulative drought duration in a year.
Drought frequency is expressed by the number of drought events during a study period. Thus, the annual
drought frequency was defined by the cumulative number of droughts within a water year (Tallaksen
and Lanen, 2004). The number of droughts was the cumulative frequency of droughts within a time
period (e.g., a year, several years). In addition, the annual minimum flows of each water year were
extracted and used to determine the model's ability to simulate very low flows. The drought volume



deficit was calculated as:
$$D_k = \sum_{t_i}^{t_j} \left( Q_{90,t} - Q_n \right) \cdot 60 \cdot 60 \cdot 24$$ (4)
where $D_k$ is the drought volume deficit (m$^3$) of a drought event $k$ at a hydrological station and $t_{k,i}$ and $t_{k,j}$
are the initial and final time steps of the run, respectively. $Q_n$ is the daily streamflow of $n$ day of the year
(1-365). The corresponding drought duration is computed as $t_j$-$t_i$ +1.
For hydrological drought events that occur relatively close in time, the inter-event time method
introduced by Zelenhasić and Salvai(1987) was used to separate events. This method defines a minimum
gap period, $t_c$, and assumes that if the inter-event time ($t_j$-$t_i$+1) < $t_c$, then the consecutive events are
interdependent and merged. In this case, the total drought deficit volume is the sum of the individual
deficit values, and the event duration is the so-called real drought duration (sum of the single event
duration, excluding excess periods). For this study, $t_c$ was set equal to 7 days as recommended by
Cammalleri et al.(2017).

**3. Results**
3.1 Model performance on daily streamflow and hydrography
Fig. 3 depicts model performances for calibration results in the presence or absence of the wetlands
and the combination of wetlands and reservoir at the ten hydrological stations in the NRB. In the case
of whether the wetlands were present or absent, the simulated daily streamflow results all achieved the
acceptable performance criteria (NSE > 0.5 and Pbias ≤±15%) suggested by Moriasi(2007) and Moriasi
et al.(2015) at the Shihuiyao, Guli, Huolengmen, Kumotun, Kehou, Liujiatun and Kumotun stations.
However, compared with the calibrated results of the model without wetlands, the simulation efficiency
under with wetland scenario improved to varying degrees. Specifically, the relative improvement (i.e.,
the relative change) of KGE values at Shihuiyao, Guli, Huolengmen, Kumotun, Kehou, Liujiatun,
Kumotun, Tongmeng, Fulaerji and Dalai were 44%, 24%, 2%, 6%, 5%, 3%, 4%, 46%, 47% and 67%,
respectively. In addition, the NSE and CC values were generally larger in the presence of wetlands than
those in the absence of wetlands, and the RMSE and Pbias values are generally smaller than those in



the absence of wetlands, showing that integrating wetlands into the hydrological model can slightly
improve the model calibration results.

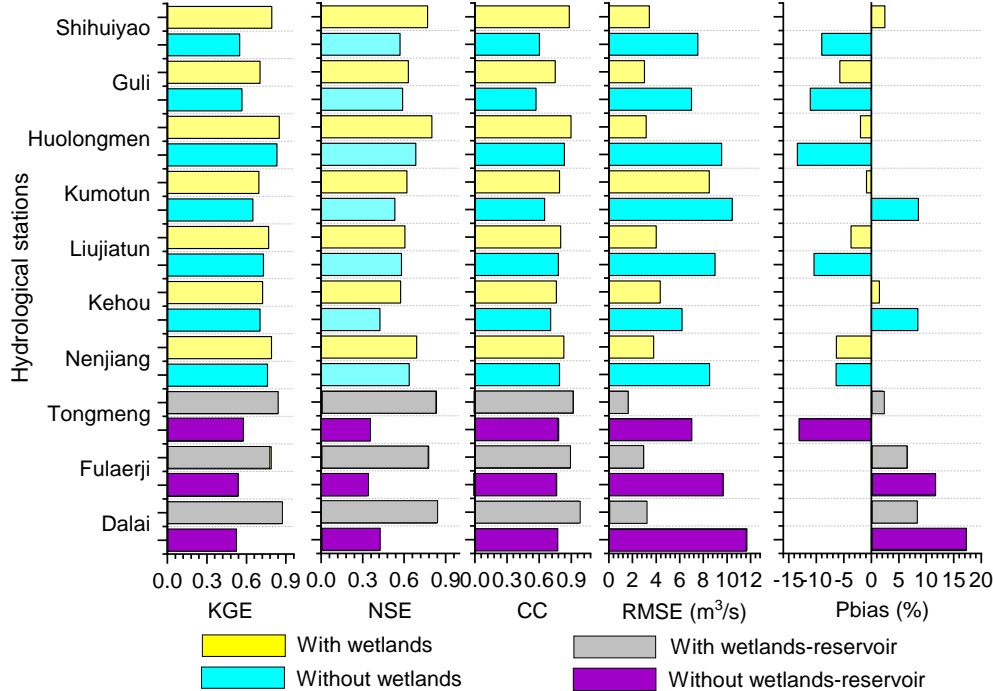


Figure 3. Model performances for calibration results for the with/without wetlands and reservoir
scenarios at the ten hydrological stations in the Nenjiang River Basin. The KGE, NSE, CC, KGE,
RMSE and Pbias refer to Kling-Gupta efficiency, Nash-Sutcliffe efficiency, Correlation Coefficient,
Root Mean Square Error, and the percentage bias, respectively.

For the lower reaches of Nierji Reservoir (i.e., the Tongmeng, Fulaerji and Dalai stations, representing
inclusion of the wetlands and the reservoir operation into hydrological modeling), the NSE and CC
values were greatly higher and RMSE and Pbias values were substantially lower when the wetlands and
reservoir were considered, in comparison to the case without wetlands-reservoir (Fig. 3). Specially, in
the scenario without wetlands-reservoir, the simulated daily streamflow results failed the acceptable
performance criteria (NSE > 0.5 and Pbias ≤±15% as suggested by Moriasi (2007) and Moriasi et al.
(2015). In addition, the simulated daily streamflow in the no-wetland and no wetlands-reservoir





scenarios both overestimated the high flows, especially those during the flood periods; during the low
flow periods, the low flows were underestimated (Please refer to Fig. A1 in Supplementary materials).
Further, the simulated hydrographs under the wetland and wetlands-reservoir scenario were in much
better agreement with the hydrographs of observed streamflow, especially during floods and the low-
flow period (Please refer to Fig. A2 in Supplementary materials). These results indicate that inclusion
of the wetlands and the operation of reservoirs can greatly improve model capacity to replicate basic
hydrograph characteristics and capture hydrological extremes (e.g., high and low flows).
3.2 Model capacity to replicate flood and drought characteristics
The simulated annual minimum streamflow for the wetlands/wetlands-reservoir scenarios were, in
general, slightly overestimated or approximately equivalent to the observations compared to the
scenarios that did not include the wetlands/wetlands-reservoir (Please refer to Fig. A3 in Supplementary
materials). However, the simulation results without wetlands clearly underestimated minimum
streamflow, distinctly overestimated annual drought days and drought deficit compared to the
simulation results for the scenario with wetlands at the ten hydrological stations. In addition, the
simulated annual maximum peak flow, flood days and volume under the with/without wetland scenarios
are, in general, approximately comparable to observations at the Guli, Kumotun, Kehou, Liujiatun and
Nenjiang hydrological stations (Please refer to Fig. A4 in Supplementary materials). Specifically, for
the upstream Nierji Reservoir, it is apparent that if wetlands are not considered, the number of annual
flood days will be overestimated, whereas flood volume will be substantially underestimate at the
Huoloengmen Station. For the lower reach of Nierji Reservoir, lack of integrating the wetlands and
reservoir into the simulation can lead to a notable underestimation of annual flood days, and a substantial
overestimation of the annual maximum peak flow and flood volume. These results demonstrate that
integrating wetlands and the combination of wetlands and the reservoir into the model can help improve
model performance with regards to flow during the calibration process, and enhances the model's
capability of depicting streamflow processes as well as capturing flood and drought characteristics.





3.3 Projection of future floods
A comparison between historical and projected flood characteristics at the Nenjiang Station
(representing inclusion of wetlands into hydrological modeling) shows an overall increase in flood risks
in the upper NRB. The flood duration, peak flow, volume and flashiness generally exhibit larger
fluctuations in most of the scenarios (different SSPs and three periods as shown in Fig.4a-d, Fig.5a-d
and Table A1). In addition, the averaged increase in flood duration, peak flow, volume and flashiness
ranges from 0.9 to 1.2%, from 16 to 33%, from 8 to 111% and from 26 to 55%, respectively (Fig.5).
Specifically, the extreme values of flood duration are much larger during the near future and end-century
under the SSP126 scenario, the end-century under the SSP370 scenario and the mid- and end-century
under the SSP585 scenario. Apart from a slight decrease during the near future and mid-century under
the SSP585 scenario, peak flow will increase through time in the SSP126, SSP370 and SSP585 scenarios
(Fig. 4b). Simultaneously, the flood volume will experience the greatest increase of 68% during the near
future under the SSP585 scenario, followed by a 22% increase during the mid-century under the SSP126
scenario. In terms of flashiness, the floods will be more severe under the constrains inherent in the
SSP126 and SSP585 scenarios and less severe given the conditions in the SSP370 scenario, as compared
to the historical period (Fig. 4d).
It should be noted that the flood duration, peak flow, volume and flashiness can decrease in the future,
as compared to the historical period (Fig.5). For example, flood duration will slightly decrease during
the near future and end-century under the SSP126 scenario, largely decrease during the near future under
the SSP585 scenario, respectively. Under the SSP585 scenario, the flood peak flow will experience a
decrease with the percentage change values of 15% during the mid-century and the volume will reduce
26% during near future. In addition, future flood flashiness will be reduced by 49% and 28% in the near
future and the end-century under the SSP370 scenario respectively, and by 21% at mid-century under
the SSP585 scenario.



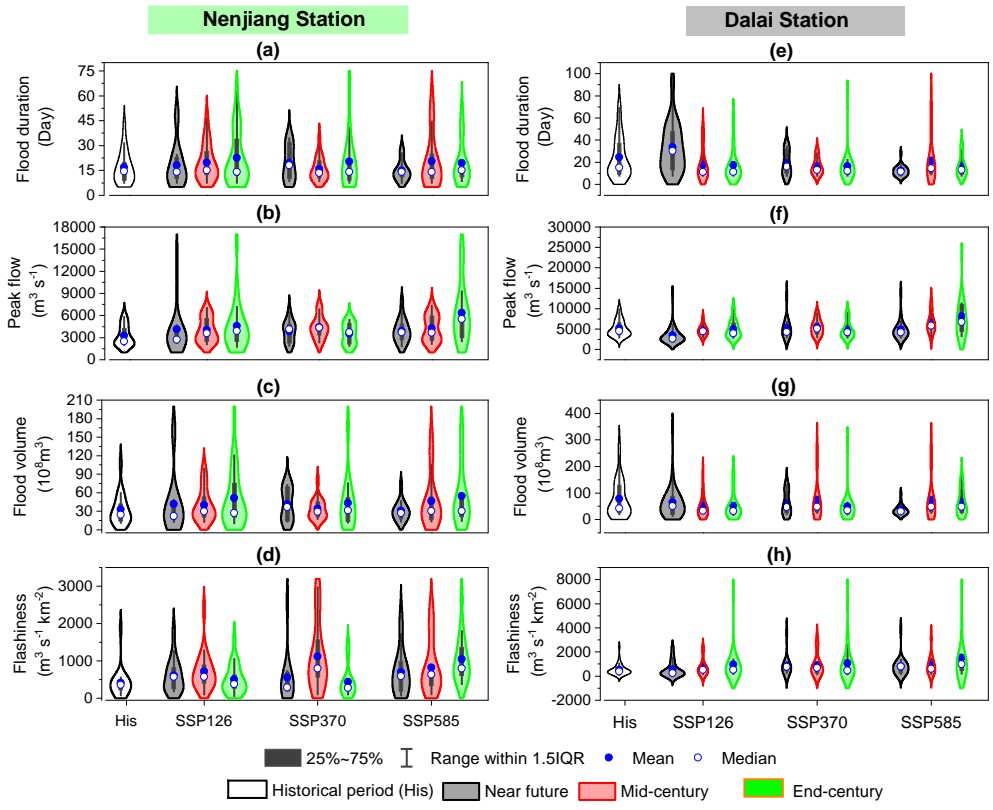


Figure 4. Historical and projected flood duration, peak flow, volume and flashiness at the Nenjiang (the

left column) and Dalai (the right column) Station. The historical period refers to 1971-2020 and the

near-future, mid-century and end-century refer to the 2026-2050, 2051-2075 and 2076-2100 under the

Socioeconomic Pathways (SSP) 126, SSP370 and SSP585 scenarios. Note that the wider the violin plot,

the higher the density.

The changes in the historical and future flood duration, peak flow, volume and flashiness at the Dalai

Station (representing inclusion of downstream wetlands and reservoir operation into hydrological

modeling) is shown in Fig.4 e-h and Fig.5 e-h. Similar to the Nenjiang station, the flood duration, peak

flow, volume and flashiness at the Dalai station also exhibit divergent change trends across different

SSPs and three periods, as compared to the historical periods. Flood duration is projected to increase

largely in the near-future period for the SSP126 scenario, both in the mid-century and end-century for





the SSP370 scenario. The peak flow will broadly decrease for the SSP126 scenario, increase for the

SSP370 and 585 scenarios. The relative change of flood volume will be varying considerably and

contrarily from near-future to the end-century. Flood volume will decrease in the near-future and

decrease in the end-century for both scenarios of SSP126 and SSP370. Flashiness will be reduced in the

near-century and will increase in the mid-century and end-century for the SSP126 scenario. For the

SSP370 scenario, flashiness will increase substantially with percentage changes of 204% in the near-

future, Moreover, for the SSP585 scenario, flashiness will experience a considerable increase of

flashiness with percentage changes of 109% in the end-century respectively. In terms of the averaged

percentage change values, the peak flow and flood flashiness will overall increase; the flood volume

will reduce in the near future and rise in the mid-century and end-century; and flood duration will

experience a slight increase to a minor decrease.

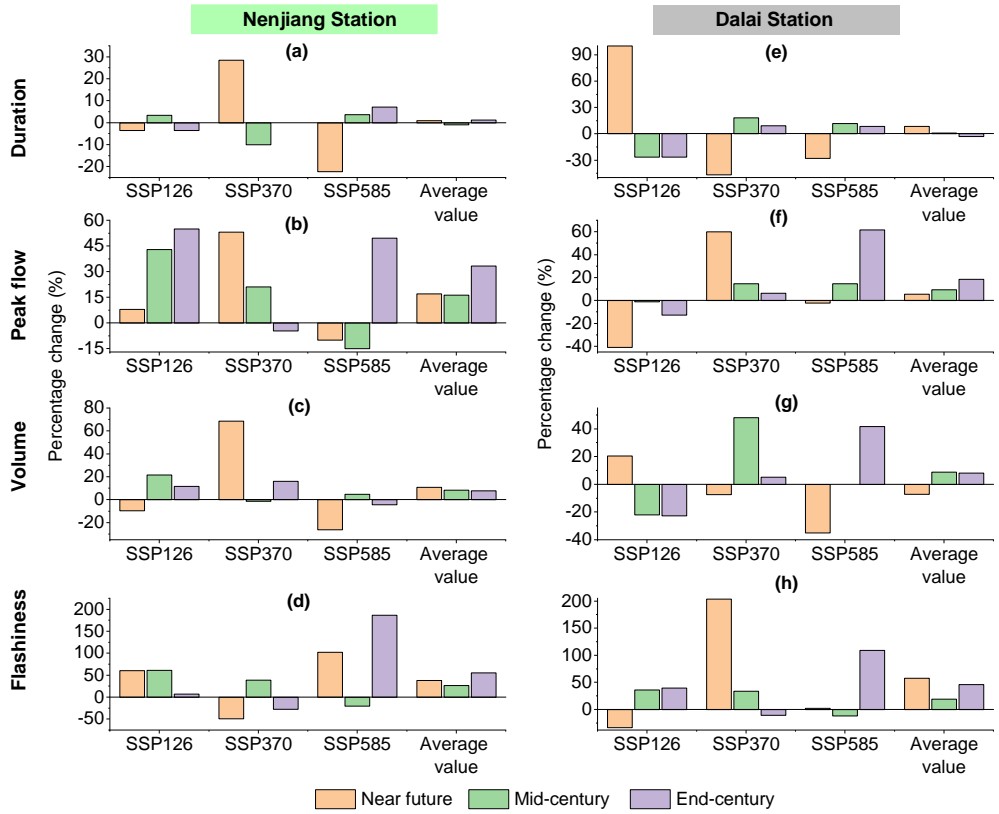





Figure 5. Projected percentage changes (relative to historical period during 1971-2020) in flood duration,
peak flow, volume and flashiness at the Nenjiang (the left column) and Dalai (the right column) Station. The
near-future, mid-century and end-century refer to the 2026-2050, 2051-2075 and 2076-2100 under the
Socioeconomic Pathways (SSP) 126, SSP370 and SSP585 scenarios. The average values were calculated
based on the projected percentage changes in the three SSP scenarios.

To further investigate flood risks in the NRB under future climate change, the flood duration-peak
flow-flow volume relationships at the Nenjiang and Dalai stations for the SSPs were compared to those
of the historical period and analyzed (Fig. 6a-c). Compared with historical flood risk, extreme flood
events with longer and larger volumes will occur more frequently at the Nenjiang Station for the SSP126
and SSP585 scenarios (Fig. 6a and 6c). It is noteworthy that the flood peak-volume-duration
relationships between the historical period and SSP370 scenario are approximate equal, with the
exception that longer duration and larger volume floods will occur during the end-century period (Fig.
6b). In addition, extreme flood events will occur mainly in the near-future for the SSP126 scenario and
during the medium and far future periods for the SSP585 scenario. Moreover, for the SSP370 and
SSP585 scenarios, floods will become shorter in duration, and possess a lower peak flow and flood
volume in the near-future. Thus, the upper NRB will experience more severe flood events to a large
extent under most future climate change.



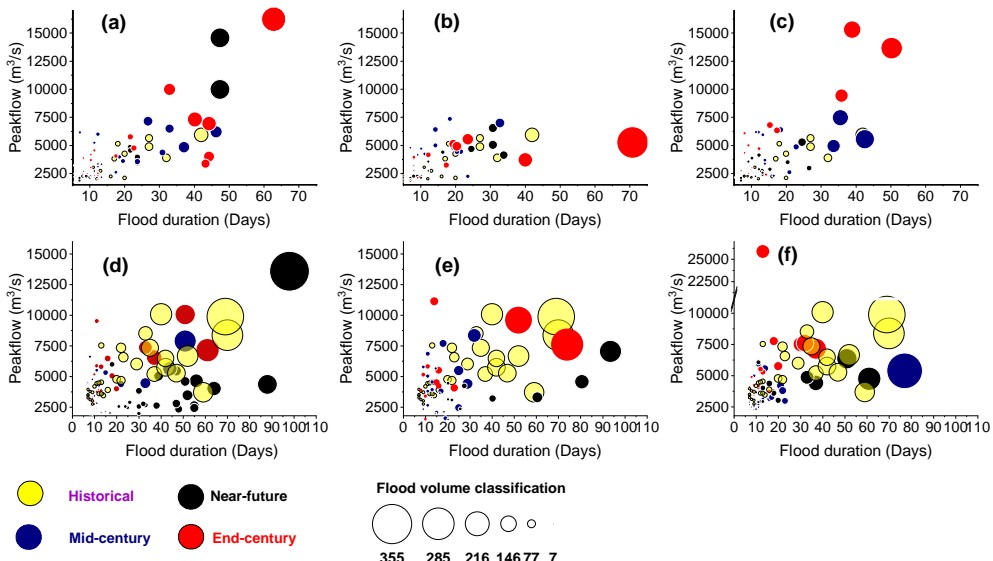

Figure 6. Historical and projected flood duration-peak flow-volume relationships at the Nenjiang (the first row) and Dalai (the second row) Station. The historical period refers to 1971-2020 and the near-future, mid-century and end-century refer to the 2026-2050, 2051-2075 and 2076-2100 under the Socioeconomic Pathways (SSP) 126, SSP370 and SSP585 scenarios.

The duration-peak flow-volume relationships of extreme flood events under future climate change scenarios are closer to those of the historical period at the Dalai Station than at the Nenjiang Station (Fig. 6e-f). For the three future SSPs, the flood events with longer duration, higher peak flows or larger volume than the historical period will occurred infrequently, and the duration, flood volume and peak flow of the other shorter and lower magnitude flood events will generally be attenuated. However, very extreme flood events are projected to occur in the near-future under the conditions of scenario SSP126. Likewise, future climate change under the SSP370 scenario and 585 scenarios are projected to result in longer flood events in the near-future and mid-century, respectively. Therefore, the future flood risk can be effectively attenuated to a great extent by the combined influence of wetlands and reservoir. However, the fact that extreme flood events that will still occur in the future.





3.4. Prediction of future hydrological droughts
The comparison between historical and projected hydrological drought indices shows that the risks
of hydrological droughts will be increase to some extent under future climate change for both Nenjiang
and Dalai stations. Specifically, in addition to a reduction in the number of droughts and annual drought
days in the near future for the SSP126 scenario, the number of droughts, annual drought days and
drought deficit will overall increase in other periods for three scenarios (Fig. 7 and Fig. 8; Table A2). It
is clearly that the number of droughts will equivalent to the historical period in the mid-century and end-
century for the SSP126 scenario and in the mid-century for the SSP585 scenario. For all other scenarios,
the number of droughts will increase. In terms of the mean percentage change values, there is a general
trend towards an increase in the number of droughts and annual drought days, which indicate that future
drought events will be more frequent and there will be more days per year affected by drought (Fig. 8).
The predicted extreme values show that the future duration of drought at Nengjiang station may shorter
than the historical period, but the degree of shorting presented in different SSP scenarios varies. For the
Dalai station, the longest drought durations would all exceed historical extremes in the end-century for
the ssp126 and SSP585 scenario, and in the near future for the SSP370 scenario. The percentage change
values display that drought duration will be reduced at the Nenjiang station and will be extended at the
Dalai station for all the SSP scenarios. Drought deficit at the Dalai station will increase by 39%, 36%
and 36% and in the near future, mid-century and end-century. For the Dalai station, drought deficit will
increase further in the three periods with 39%, 36% and 36%, respectively.
A comparison of the percentage change values between the Nengjiang and Dalai stations shows that,
apart from a reduction of the number of drought events, the risk of drought to be experienced at Dalai
is considerably stronger than at Nengjiang. Specifically, the percentage change in the annual drought
days, drought duration and deficit will increase from 85-97% to 89-134%, from -17- -17% to 21%, and
from 36-39% to 171-247%, respectively.

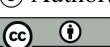


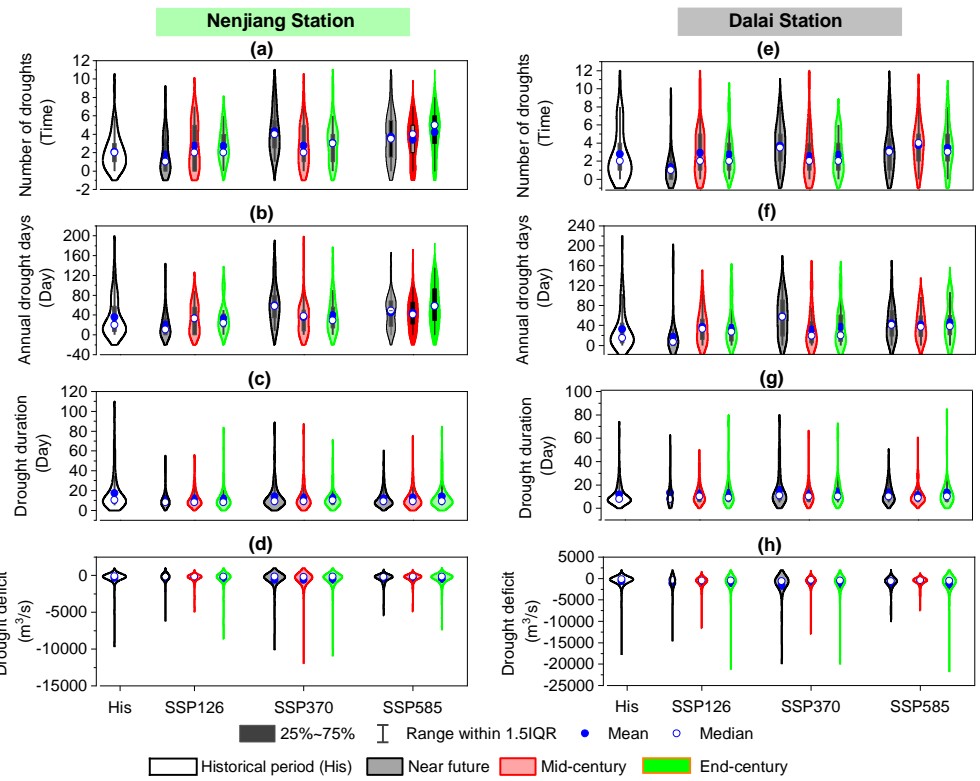


Figure 7. Historical and projected hydrological drought characteristics (the number of droughts, annual

drought days, duration, and deficit) at the Nenjiang (the left column) and Dalai (the right column) Station.

The historical period refers to 1971-2020 and the near-future, mid-century and end-century refer to the 2026-

2050, 2051-2075 and 2076-2100 under the Socioeconomic Pathways (SSP) 126, SSP370 and SSP585

scenarios. Note that the wider the violin plot, the higher the density.





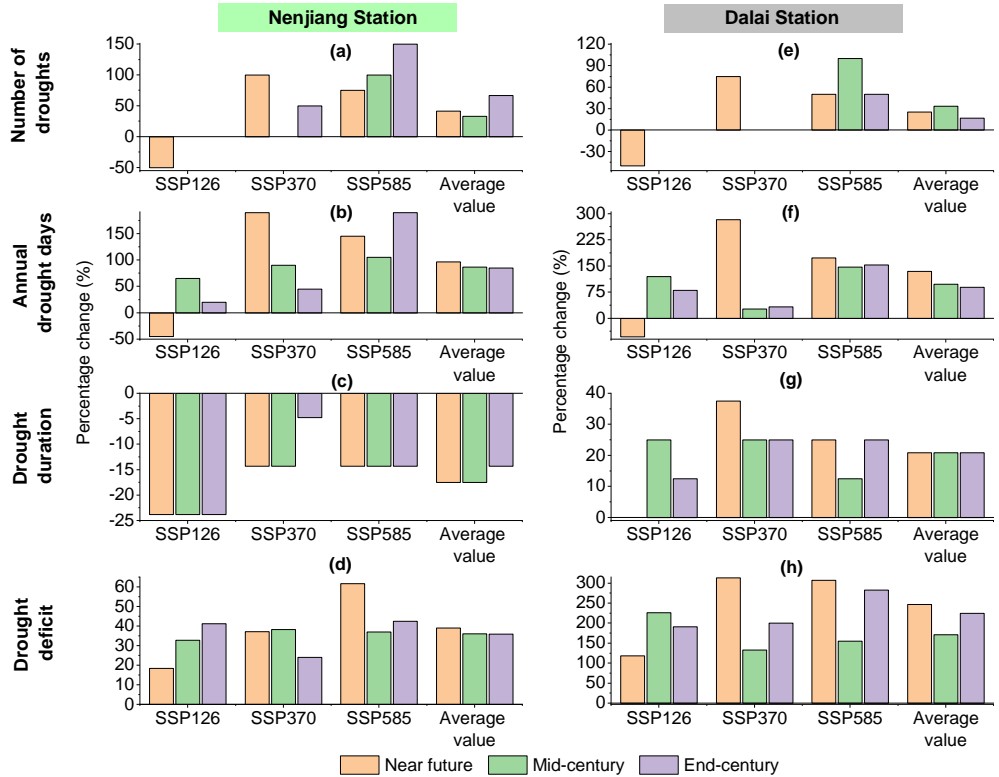

Figure 8. Projected percentage changes (relative to historical period during 1971-2020) in hydrological drought characteristics at the Nenjiang (the left column) and Dalai (the right column) Station. The near-future, mid-century and end-century refer to the 2026-2050, 2051-2075 and 2076-2100 under the Socioeconomic Pathways (SSP) 126, SSP370 and SSP585 scenarios. The average values were calculated based on the projected percentage changes in the three SSP scenarios.

To further analyze the temporal evolution of droughts in the Nengjiang River Basin under future climate change, drought events were classified into four types in terms of duration and deficit, i.e., short-term light droughts, long-term light droughts, short-term severe droughts, and long-term severe droughts (see Fig. 9 for details). This four-part classification was then used to compare and analyze the changes in the temporal characteristics of drought events under the different SSP scenarios. Similar to the drought characteristics during the historic historical period, the majority of drought events for the



SSP126, SSP370 and SSP585 scenarios are short-term light droughts (Fig. 9a, 9b and 9c), i.e., the upper
NRB will still be dominated by short-term light droughts under future climate change. However, these
droughts will be slightly aggravated and marginally longer. In addition, long-term light droughts will
occur rarely under the conditions inherent in scenarios SSP126 and SSP370, and occur relative
frequently in the SSP585 scenario. However, compared with the historical period, the overall number
of long-term light droughts will largely decrease, but the deficit will increase slightly under future
climate change. In addition, short-term severe droughts will increase substantially, along with their
deficit. The number of long-term severe droughts for the SSP126 scenario is approximately the same as
in the past, but the duration will be substantially reduced. For scenarios SSP370 and SSP585, the number
of long-term severe droughts will increase more than during the historical period, but the duration will
be markedly less, and the deficit will be reduced to some extent. In terms of the different the sub-periods,
severe droughts in the upper NRB will be more severe during the near-future and end-century periods,
and relatively less severe in the mid-century period in comparison to the historical period. However,
overall, the droughts will be of shorter duration and characterized by an increased deficit under future
climates.

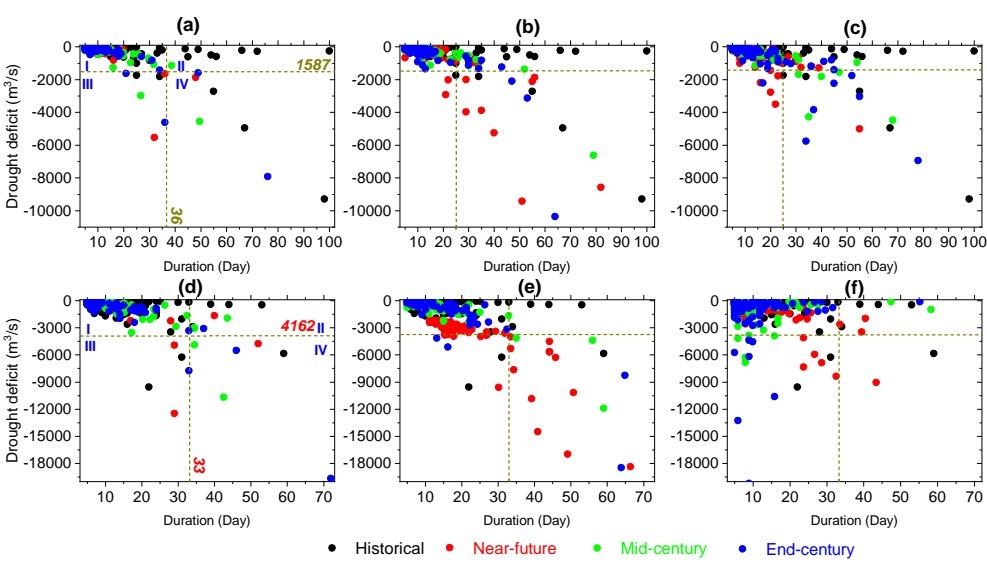




Figure 9. Historical and projected duration-deficit relationship of each hydrological droughts at the Nenjiang (the first row) and Dalai (the second row) Station. The historical period refers to 1971-2020 and the near-future, mid-century and end-century refer to the 2026-2050, 2051-2075 and 2076-2100 under the Socioeconomic Pathways (SSP) 126, SSP370 and SSP585 scenarios. The dark yellow lines in the horizontal and vertical directions refer the 95% threshold lines for drought deficit and duration values, respectively. I, II, III and IV refer to short-term light droughts, long-term light droughts, short-term severe droughts, and long-term severe droughts, respectively.

Droughts brought about by future climate change at the Dalai Station located along the lower reaches of the NRB will continue to be dominated by short-term slight droughts (Fig. 9d, 9e and 9f). For the SSP126 scenario, the duration and deficit of the short-term slight droughts will be approximately the same as those during historical times. However, the duration and deficit of short-term slight droughts will increase given the conditions specified in the SSP370 and SSP585 scenarios. The duration of short-term slight droughts will increase the most for scenario SSP370. In addition, under all three SSP scenarios, long-term slight droughts will, in general, be reduced. In fact, under the SSP370 scenario, long-term slight droughts will not occur. The number of short-term severe droughts will generally tend to increase, with the most pronounced increase under the SSP585 scenario, followed by the SSP370 scenario. A slight increase will occur under the SSP126 scenario. However, long-term severe droughts will increase substantially under the SSP126 and SSP370 scenarios. In particular, under the SSP370 scenario, the duration of long-term severe droughts will be exceptionally prolonged, and the severity will be extraordinarily increased, indicating that the risk of droughts of long duration and with a severe deficit will climb abnormally in some year. For example, under the conditions set by the SSP370 scenario, the deficit of long-term severe droughts will reach -18,169 $m^3$ and -18,457 $m^3$ during the near-future and end-century periods. For the SSP585 scenario, long-term severe drought will occur only once in the near-future, which is equivalent to the historical period. These results indicate that the risk of future hydrologic droughts along the lower NRB will further increase even under the combined influence of reservoirs and wetlands.





**4. Discussion**

4.1. Integrating wetlands and reservoir operation into basin hydrologic modeling and basin water management

A series of studies have shown that the simulation and prediction of floods and droughts faces many challenges, such as the scarcity of hydrometeorological driven data (Foulon et al., 2018), model errors (Smakhtin, 2001; Staudinger et al., 2011; Golden et al., 2021) and anthropogenic disturbances (e.g., reservoir operation) (Brunner, 2021; Brunner et al., 2021). In this study, we developed a spatially explicit hydrological model that considers wetland hydrological processes and reservoir operations through coupling a distributed hydrological modeling platform with wetland modules and reservoir simulation algorithms. We found that coupling wetland alone or coupling wetlands and reservoir with hydrological model can improve model calibration results and model performance of capturing flood and drought characteristics in a large river basin. Such model performance improvement can minimize uncertainties (Zhao et al., 2016; Rajib et al., 2020b; Golden et al., 2021) and provide important information for developing downstream water resources management. Previous studies have shown that climate change is further exacerbating the risk of hydrological extremes, leading to an expanding of flood and drought affected area (e.g., Hirabayashi et al., 2013; Diffenbaugh et al., 2015b; Wang et al., 2021), which increase the complexity of accurate prediction and the challenge for effective mitigation. Give that, projecting flood and drought risks in response to a changing climate requires robust hydrologic models that take into account the important factors within a watershed that can largely influence basin hydrological processes (Golden et al., 2021).Therefore, in basins that coexist with high-coverage wetlands and multiple reservoirs, it is necessary to integrate wetlands and reservoir operation into basin hydrological simulation, thus providing practical support for extreme hydrological risk mitigation and water resource management under a changing climate.

4.2. Future flood and drought risks under the influences of upstream wetlands

We found that the risks of floods and droughts will overall increase in the upper NRB under future





climate change, even when considering flow regulation services provided by wetlands. The overall
increasing flood peak, flood volume and flashiness inform that future flood risk will be much higher
compared to the historical periods. In particular, the NRB may experience flood events with much longer
duration, extreme high peak flows, exceeding lager flood volumes and extraordinary strong flashiness
(Fig. Fig.4 a-d, Fig. 5 a-d and Table A1). These extreme floods may pose a greater risk than that caused
by the Great Flood of 1998 (a 100-year flood that resulted in huge losses in the NRB) during the
historical period. As an example, a flood events spanning 66 days with the peak flow of 16213 $m^3$/s and
with the volume of $190 \times 10^8 m^3$ will be happen during the end-century periods given the constraints of
the SSP-126 scenario (Fig.5 a-c). The increasing flood risks could be largely attribute to the increasing
precipitation extremes in the NRB under future climate change. As Wu et al.(2022) who predicted future
precipitation extremes and flood events and concluded that the increasing precipitation extremes will
bring about higher flood risks with the increase in warming levels. In addition, upstream wetlands may
aid in enhancing flood risks because wetlands in headwater areas generally tend to increase downstream
flood risks (Acreman and Holden, 2013; Wu et al., 2020; Acreman et al., 2021). Moreover, the
effectiveness of wetlands is probably limited to smaller flood and drought events (Vojinovic et al., 2021).
Therefore, upstream wetlands can't fully mitigate future flood risks under future climate change,
confirming that wetlands have limited potential to efficiently mitigate future climate change.
We also found that the duration of droughts would become longer and the number of drought days
per year will increase, accompanied by an increase in drought duration and drought deficit (Fig. 7, Fig.
8 and Table A2). Further, severe droughts in the upper NRB will be more severe under the SSP370 and
SSP585 scenarios. It is worth noting that the baseflow support function of the downstream wetlands
may help the reservoir to reduce drought risk to some extent (Min et al., 2010; Fossey and Rousseau,
2016; Ameli and Creed, 2019; Golden et al., 2021). However, when experiencing extreme droughts,
this baseflow support function of downstream wetlands remain minimal, this can be corroborated by the
substantial increase in long-term severe droughts at the Dalai Station. This is because the increased
evapotranspiration during extreme drought period can cause temporary water deficits in wetlands,
causing them to reduce low flows instead of supporting them (Bullock and Acreman, 2003; Golden et





al., 2016).
Headwater or upper wetlands remain an important part of the watershed landscape, and are often
important natural reserves that cannot be reconstructed and transformed in the same way as downstream
wetlands (Colvin et al., 2019; Acreman et al., 2021). Therefore, from perspective of NBS, their
implication in enhancing basin resilience to water hazards needs an extensively assessed. While
numerous studies have showed that flood and drought risks may increase in magnitude and frequency
(Roudier et al., 2016; Cook et al., 2020; Tabari et al., 2021), the projected results come with some degree
of uncertainty. However, this study further highlights that the amplifying effect of wetlands on flooding
cannot be ignored in the headwater areas or upper reaches of basins where wetlands are widely
distributed. It is therefore reasonable to argue that without consideration of wetlands can lead to very
different distinguished results from the actual ones, or even lead to poor decision making and probably
disaster occurrence.
4.3. The combining mitigation efficiency of wetlands and reservoir operation
The relative changes (compared with historical periods) of future flood and drought indices (Fig. 4
and Fig. 7), duration-peak flow-volume relationships (Fig. 6) and duration-deficit relationship (Fig. 9)
differ between the Nenjiang and Dalai stations under the same SSP scenario or in the same period,
indicating that reservoirs and downstream wetlands can modify the continuous propagation of upstream
flood and hydrological drought risks to the downstream. First, reservoirs and downstream wetlands can
help to reduce the risks of future floods and droughts to some extent, namely partially reduce flood peak
flow and flashiness, and decrease the number of droughts, annual drought days and drought deficit.
Second, reservoirs and downstream wetlands cannot completely eliminate flood and drought risks.
Because the flood duration and volume will overall increase at the Dalai station, especially that the
extreme floods will be more frequent in the future (Fig.6). Further, in addition to the number of droughts,
the percentage change values of the annual drought days, drought duration and deficit relative change
at the Dalai Station are greater than those of Nenjiang Station (Fig.8). This imply that the mitigation
effects on hydrological droughts is minimal. Such findings suggest that future climate change will lead
to an increase in the risk of hydrologic failure of existing basic grey (e.g., reservoir) and green (i.e.,





wetlands) infrastructures, thus posing large challenges for future socio-and eco-hydrological systems in
the downstream NRB.
4.4 Implications for flood and drought risk management under climate change

This modeling study predicts higher flood and drought risks in the NRB. This could impose a great

challenge to the operation of the Nierji Reservoir dam, i.e., to its effective operation for flood mitigation
and drought alleviation. To curb the flood and drought risks caused by future climate change in the NRB,
it is urgent to improve the water regulation capacity of the lower NRB. Although the Nierji Reservoir,
as previously argued, plays an important role in reducing floods and droughts, the potential for extreme
hydrological events in the future necessitate the application of various combinations of measures with
different scales of implementation (i.e., hybrid measures) (Vojinovic et al., 2021). However, compared
with the historical period, the existing wetlands in the NRB have been seriously degraded, such as the
weakening of the connectivity between riparian wetlands and the river channel, and the increased
fragmentation of wetlands, among other changes (Chen et al., 2021). These degraded wetlands cannot
play an effective role in mitigating floods and droughts under future climate change. Therefore, we insist
that the first remedial measure to be undertaken should be the implementation of wetland restoration
and protection projects, because studies have demonstrated that wetland coverage and their spatial
pattern can affect both basin physical conditions and human decision-making attitudes toward risk
(Zedler and Kercher, 2005; Javaheri and Babbar-Sebens, 2014; Martinez-Martinez et al., 2014; Gómez-
Baggethun et al., 2019). Given that the spatial location of wetlands within a river basin is also important
in determining the efficiency of their mitigation services (Zhang and Song, 2014; Gourevitch et al.,
2020; Li et al., 2021), optimization of wetland spatial patterns should be considered and can be carried
out to further enhance the role of wetlands in flood and drought defense.

In our view, the second important remedial measure that should be implemented is to improve the

existing reservoir operation schemes based on accurate hydrological forecasting. This requires, on one
hand, coupling of wetlands with hydrological processes and models to improve the simulation accuracy
of the upstream incoming water (i.e., runoff from the Nenjiang Station) to provide scientific support for
reservoir operation decisions. Concomitantly, it is necessary to modify the existing schemes for optimal





reservoir operation to improve the system's capacity to deal with extreme flood and drought risks.
Because the percentage increase in drought (Fig. 5) and flood indicators (Fig. 8) demonstrated that the
existing reservoir operation schemes are not effective in mitigating the risks associated with future
climate change-induced floods and droughts. Therefore, we need to re-examine and evaluate the flood
and drought risks in the NRB under future climate change and propose optimal operation schemes that
can maximize the reduction of flood and drought risks by the Nierji Reservoir. Traditionally, the water
level of a reservoir should be maintained at the designed flood limited water level during the flood
season, which does not consider river flow forecast. Ding et al.(2015) analyzed a concept that provides
a dynamic control of the maximum allowed water level during the flood season, for the Nierji Reservoir
dam. A reasonable approach to tackle this issue could be to considerate forecast uncertainty and
acceptable flood risk to minimize the total loss caused by flood and drought. Further modeling studies
with multi-objective optimization algorithm can help identify an optimum reservoir operation for best
economic and ecological outcomes.
4.5. Limitation and future work

Although our developed framework demonstrates good modeling results, uncertainties could exist in

the assessment. Aspects such as the accuracy and error of the input data (Lobligeois et al., 2014), the
choice of the objective function (Fowler et al., 2018), the length of the period considered during
calibration (Arsenault et al., 2018), and the model structure (Melsen et al., 2019) can all affect the
performance of a model to replicate streamflow, thus impacting flood and drought predictions of under
future climate change. For example, the resolution of the utilized DEM affects the determination of
wetland drainage watersheds and the wetland fill-spill processes (Grimm and Chu, 2020; Zeng et al.,
2020), which in turn affects the prediction of watershed-scale surface runoff and river flow. Although
we used relatively coarse resolution data (i.e., a 1 km resolution DEM and land-use cover data) and a
short calibration period (2011-2018), we clearly demonstrate that coupling wetlands and reservoirs are
amenable to runoff simulations of high accuracy and low uncertainty. In addition, due to a lack of
wetlands water balance monitoring data, this study only used river station data (which only considered
the cumulative hydrologic effect of upstream wetlands) for model calibration. As Driscoll et al.(2020)



and Evenson et al.(2018) reported that remotely sensed inundation data can help calibrate and verify
wetland-integrated into watershed models. Therefore, there are ongoing efforts to obtain sufficient data
on wetland area dynamics and evapotranspiration, water depth and volume, soil water content using
multi-source remote sensing data and actual observations to better calibrate/validate watershed
hydrological models, which are expected to further improve the model's capacity of capturing flood and
drought patterns. Further, the distance between the Nierji reservoir and the downstream Dalai station is
535.8 km. The confluence of tributaries in the river section between the reservoir and the Dalai
hydrological station can diminish the impacts of the reservoir on floods and droughts to some extent.
Therefore, a potential limitation or bias to mention in our work is that our results may potentially
underestimate the role of reservoir operation in conjunction with wetlands.
**5. Concluding remarks**
This study explored the integrative capability of wetlands and reservoir operations in mitigating
floods and hydrological droughts under future climate change. To achieve this, we developed a modeling
framework coupling wetlands and reservoir operations into a spatially-explicit hydrological model and
then applied it in a case study involving a 297,000-km$^2$ large river basin in northeast China. With this
framework we projected future floods and hydrological droughts under five future climate change
scenarios. We found that coupling wetlands and reservoir operations can slightly increase model
calibration results and efficiently improve model capacity to capture both flood and hydrological
drought characteristics in a river basin. The upper NRB will experience more severe flood and
hydrological droughts and can impose a great challenge to the effective operation of downstream
reservoir under the predicted future climate change scenarios. The risk of future floods and hydrologic
droughts along the lower NRB will further increase even under the combined influence of reservoirs
and wetlands. These results demonstrated that the risk of floods and droughts will overall increase
further under future climate change even under the combined influence of reservoirs and wetlands,
showing the urgency to implement wetland restoration and develop accurate forecasting systems. To
fully understand how wetland and reservoir operations may be influential and maintain an acceptable

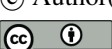



level of risk, it is therefore necessary to consider an optimization of wetland spatial patterns and
reservoir operations simultaneously, thus achieving a collaborative optimization management to
maximum basin resilience to floods and hydrological droughts. Further, the effects of combining nature-
based solutions (e.g., wetlands) with traditional engineering solutions (e.g., reservoir) should both be
useful and necessary in the future for management decisions.



**Data Availability**

The data used in this study are openly available for research purposes. The five GCM outputs (GFDL-

ESM4, IPSL-CM6A-LR, MPI-ESM1-2-HR, MRI-ESM2-0, and UKESM1-0-L) used in this study are
publically available and were provided by the Inter-Sectoral Impact Model Intercomparison Project
(ISIMIP) (https://esg.pik-potsdam.de/search/isimip/). The CMhyd software is available at
https://swat.tamu.edu/software/cmhyd. The land-use/land-cover types, soil texture, and digital elevation
model for China can be downloaded from https://www.resdc.cn/. Data from the 88 weather stations
administered by National Meteorological Information Centre of China can be download at
http://data.cma.cn.





**Appendices:**

Table A1. Median values of historical (His) and projected flood characteristics (duration, peak flow, volume, and flashiness) at the Nenjiang and Dalai stations under different Socioeconomic Pathways (SSP) scenarios in the near-future (IV), mid-century (II) and end-century (III).

| | Duration | | | Peak flow (m³/s) | | | Volume (m³) | | | Flashiness | | |
|---|---|---|---|---|---|---|---|---|---|---|---|---|
| | SSP126 | SSP370 | SSP585 | SSP126 | SSP370 | SSP585 | SSP126 | SSP370 | SSP585 | SSP126 | SSP370 | SSP585 |
| *Nenjiang* | | | | | | | | | | | | |
| His | 14.5 | 14.5 | 14.5 | 2499.2 | 2499.2 | 2499.2 | 24.1 | 24.1 | 24.1 | 356.5 | 356.5 | 356.5 |
| I | 14 | 15 | 14 | 2697.8 | 3570.8 | 3872.6 | 21.8 | 29.34 | 26.9 | 571.6 | 575.5 | 382.4 |
| II | 18 | 13.5 | 14 | 4133.1 | 4326.5 | 3693.9 | 36.8 | 28.9 | 31.3 | 290.2 | 797.5 | 276.9 |
| III | 14 | 14 | 15 | 3724.8 | 3678.5 | 5525.6 | 27.1 | 30.3 | 29.9 | 588.5 | 633.7 | 793.2 |
| *Dalai* | | | | | | | | | | | | |
| His | 15 | 15 | 15 | 4498.9 | 4498.9 | 4498.9 | 41.7 | 41.7 | 41.7 | 377 | 377.1 | 377.1 |
| I | 30 | 11 | 11 | 2661.3 | 4450.9 | 3925.2 | 50.3 | 32.6 | 32.3 | 251.3 | 513.1 | 527.7 |
| II | 16 | 13 | 12 | 4259.3 | 5111.1 | 4179.3 | 46.7 | 48.3 | 33.9 | 763 | 686.2 | 473.2 |
| III | 11.5 | 14.5 | 13 | 4171.8 | 5859.3 | 6762.5 | 30.4 | 48.3 | 48.1 | 781.4 | 605.1 | 990.4 |

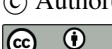



Table A2. Median values of historical (His) and projected of the projected number of droughts, annual
drought days, duration, and deficit of each drought at the Nenjiang and Dalai stations under different
Socioeconomic Pathways (SSP) scenarios in the near-future (I), mid-century (II) and end-century (III).

| | Number of droughts | | | Annual drought days (days) | | | Drought duration (days) | | | Drought deficit (m$^3$) | | |
|---|---|---|---|---|---|---|---|---|---|---|---|---|
| | SSP126 | SSP370 | SSP585 | SSP126 | SSP370 | SSP585 | SSP126 | SSP370 | SSP585 | SSP126 | SSP370 | SSP585 |
| *Nenjiang* | | | | | | | | | | | | |
| His | 2 | 2 | 2 | 20 | 20 | 20 | 10.5 | 10.5 | 10.5 | -115.6 | -115.6 | -115.6 |
| I | 1 | 2 | 2 | 11 | 33 | 24 | 8 | 8 | 8 | -136.8 | -153.5 | -163.3 |
| II | 4 | 2 | 3 | 58 | 38 | 29 | 9 | 9 | 10 | -158.5 | -159.8 | -143.4 |
| III | 3.5 | 4 | 5 | 49 | 41 | 58 | 9 | 9 | 9 | -186.7 | -158.3 | -164.6 |
| *Dalai* | | | | | | | | | | | | |
| His | 2 | 2 | 2 | 15 | 15 | 15 | 8 | 8 | 8 | -154.4 | -154.4 | -154.4 |
| I | 1 | 2 | 2 | 7 | 33 | 27 | 8 | 10 | 9 | -337.6 | -503.4 | -449.1 |
| II | 3.5 | 2 | 2 | 57.5 | 19 | 20 | 11 | 10 | 10 | -639.1 | -360 | -464 |
| III | 3 | 4 | 3 | 41 | 37 | 38 | 10 | 9 | 10 | -629.1 | -394.2 | -591 |


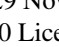


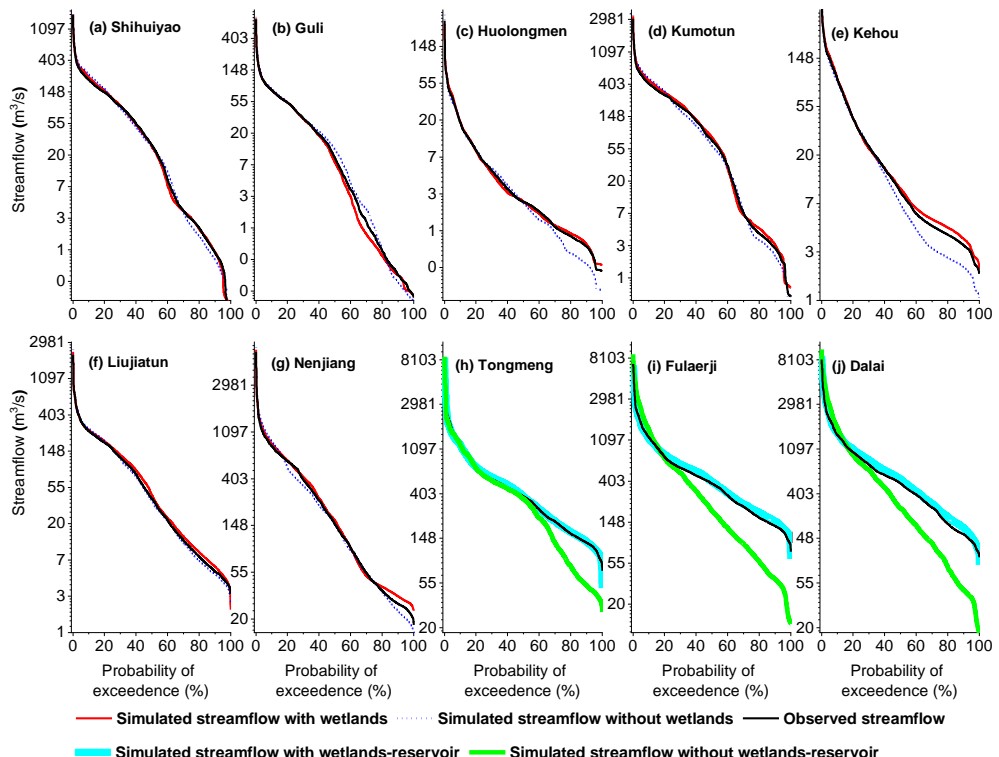


Fig. A1. Comparison of daily flow duration curves at ten hydrological stations in the Nenjiang River Basin.
The simulated streamflow used in Fig. 7a-g were calibrated with/without wetlands whereas the simulated
streamflow used in Fig. 7h-j were calibrated with/without wetlands and the Nierji reservoir.




Fig. A2. Comparison of daily simulated and observed streamflow at ten hydrological stations in the Nenjiang

River Basin. The simulated streamflow used in Fig. 6a-g were calibrated with/without wetlands whereas the

simulated streamflow used in Fig. 6h-j were calibrated with/without wetlands and the reservoir.



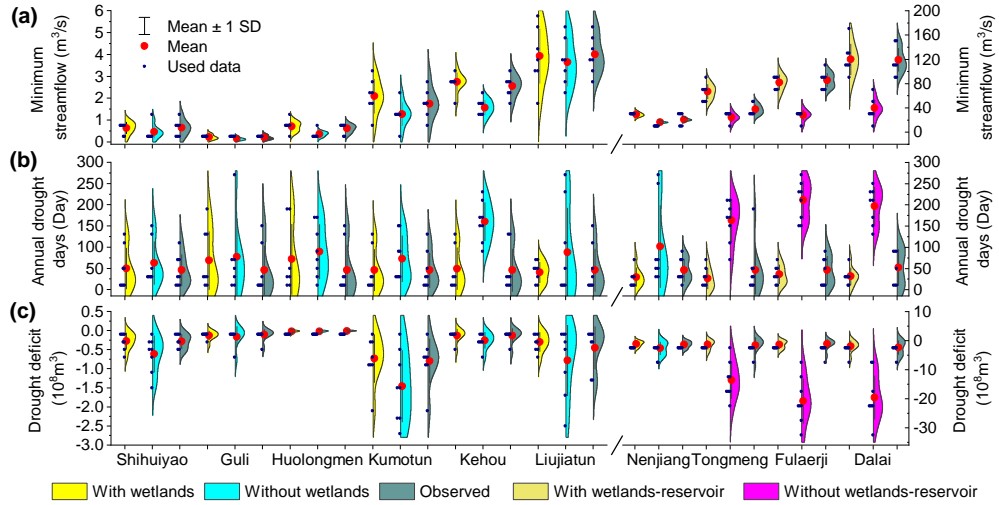

Fig. A3. Annual minimum streamflow, drought days and deficit derived from observed records and simulated

streamflow at ten hydrological stations in the Nenjiang River Basin. The with and without wetlands/wetlands-

reservoir refers to streamflow simulation based on the presence or absence of wetlands/wetlands and reservoir.





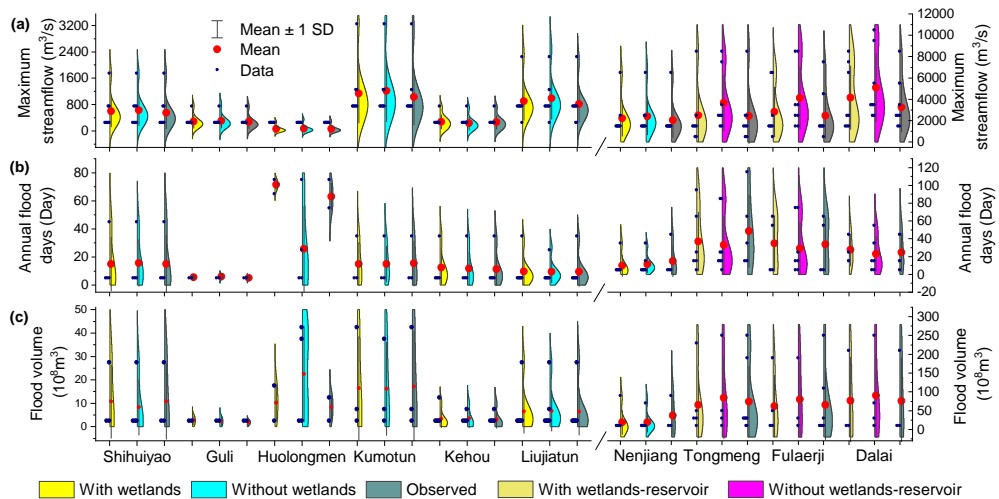

Fig. A4. Annual maximum peak flow, flood days and volume derived from observed records and simulated

streamflow at ten hydrological stations in the Nenjiang River Basin. The with and without

wetlands/wetlands-reservoir refers to streamflow simulation based on the presence or absence of

wetlands/wetlands and reservoir.

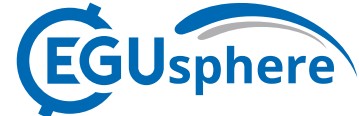

**Author contribution**
**Yanfeng Wu**: Conceptualization, Writing, Data analysis, Methodology, Software. **Jingxuan Sun**:
Formal analysis, Investigation, Data analysis and Plotting. **Boting Hu**: Software, Visualization, Data
analysis. **Y. Jun Xu**: Writing - review &editing. **Alain N. Rousseau**: Writing - review & editing.
**Guangxin Zhang**: Conceptualization, Supervision, review &editing.

**Competing interests:**
The authors declare that they have no known competing financial interests or personal relationships
that could have appeared to influence the work reported in this paper.

**Acknowledgments**
This work was supported by the National Natural Science Foundation of China (42101051 and
41877160), the Postdoctoral Science Foundation of China (2021M693155), the Strategic Priority
Research Program of the Chinese Academy of Sciences, China (XDA28020501, XDA28020105), and
The National Key Research and Development Program of China (2021YFC3200203). During
preparation of this manuscript, YJX received a grant from U.S. Department of Agriculture Hatch Fund
(project number, LAB94459).

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
