# Peer review of "Can the combining of wetlands with reservoir operation reduce the risk of future"

_EGUsphere, 2022_

## Author Comment (AC3)

**Response to Anonymous Referee #3**

**General comments:**

The paper entitled "Can the combining of wetlands with reservoir operation largely reduce the risk of future flood and droughts?" is an interesting work that integrates wetlands as nature-based solution for mitigation of flood and droughts in modelling. This work helps improve the understanding of how to integrate wetlands into hydrological models as well as how wetlands can be used for hydrological assessments. I believe it deserves publishing but with major revisions.

This paper is a long and sometimes repetitive read in which it is easy to lose track of all the given information. Overall, the article could be more condensed and straightforward. Moreover, some clarification concerning the modeling approach, more explicit description of the methods as well as better connection to the aim and research questions is needed. For instance, the work of future projection of flood and droughts are given more space than could be understood in the aim and research questions and also compare to what the reader initially would think regarding the focus on wetlands with reservoir operations as mentioned in the title. It is not clear how this study comes to the conclusions regarding how wetland and reservoir operation reduces the risk of future floods and droughts. I believe this should be the main aspects of the results and discussion. In addition, the results and discussion should better reflect the fact that this work is based on a case study and that local conditions/mitigations from wetlands combined with reservoir operation could vary between river basins.

> Response: Thank you very much for your careful assessment of our manuscript and for acknowledging the value of our work. We highly appreciate your constructive comments and have diligently revised the manuscript. Please find our responses to the detailed comments below.

**Specific comments:**

Introduction- very long and repetitive introduction that can be condensed to the most important and relevant aspects of the study.

> Response: We thank your comment here. Mostly based on your constructive comments, and taking into account the suggestions of Referee #1 and Referee #2, we condensed the Introduction to preserve the most important and relevant aspects of the study.

Line 47: what do you mean by cascade up the flood risks to a great extent?

> Response: We apologize for unclear representation here. We have changed 'cascade up' to 'mitigate' in the sentence as follows:
>
> Concurrently, the disaster-related loss of ecosystems (e.g., wetlands, forest and grassland) and their services can mitigate the flood and drought risks to a great extent (Gulbin et al., 2019; Walz et al., 2021).

Line 53-68 if you are referring to wetlands as NBS, why not write it in a same unifying paragraph?

Response: We thank your comment here. We have carefully checked and revised the full text to make it in a same unifying paragraph.

Line 74-75 what do you mean by hydrologic equivalent wetland.

Response: We thank your comment here. Due to our drastic changes to Introduction, the hydrologic equivalent wetland has been removed here. But in the following text, we describe the hydrologic equivalent wetland in detail as follows:

It should be mentioned that the HEW concept developed by Wang et al (2008) served as the foundation for the integration of RWs and IWs into the modeling framework. This concept contends that the features of one HEW (also known as an isolated wetland or riparian wetland) are equivalent to the sum of the characteristics of each wetland inside a RHHU (which could either be hill slopes or elementary sub-watersheds related to one river segment). The following premises apply to this concept: (i) only one isolated and/or riparian HEW per RHHU; (ii) one HEW can be fully integrated within a RHHU; (iii) isolated HEW parameters must be numerically integrated; and (iv) riparian HEW parameters must be numerically integrated and spatially integrated (i.e., located in a specific location on the river segment). Therefore, IWs and RWs do not appear to have direct hydrological connection within a RHHU. However, IWs also have hydrological interactions with RWs through vertical water balance processes and fill-spill processes (Fossey et al., 2015).

Line 76-77: instead of typing all the references, you could shorten and simplify the reading to " multiple studies (e.g., references)"

Response: We agree with your opinion here and have rephrased the sentence as follows:

Since then, multiple studies (e.g., Evenson et al., 2016; Evenson et al., 2018; Zeng et al., 2020) successively modified wetland modules (isolated or riparian wetlands) and improved the applicability of SWAT model to discern hydrological services of basin wetlands.

Line 93-95: sentences that could be rewritten into previous sentence inline 91-93

Response: We agree with your comment here and have rephrased the two sentence as follows:

Such numerous reservoirs and their large storage capacity should not be neglected in water hazard assessment and hydrological projection because of their significant modification on flood and drought patterns (Boulange et al., 2021; Brunner et al., 2021).

Line 95-96: how is this information relevant to the study?

Response: We agree with you that this sentence is not relevant to the study and we have deleted the sentence.

Line 115: what is meant by the expression " 1+1=2" simulation effect?

Response: Thanks for your kind reminder here. This expression in unclear and we have rephrased the sentences as follows:

Furthermore, on a global scale, most river basins have wetlands and their river flow has or will experience reservoir regulation (Schneider et al., 2017; Muller, 2019), which elicits a thought-provoking concern: What will be the changes of future floods and droughts under the combined influence of wetlands and reservoirs?

Line: 137: I think it is worth mentioning and clarifying that you use the Nenjiang river basin as a case study to answer the aim of the effect of wetland and reservoir operations into hydrological modelling for mitigating future flood and drought.

Response: We thank your comment here and have added the description about this point as follows:

The Nejinang River Basin was selected as a case study here because it has abundant wetlands and a large reservoir, and has undergone intensive anthropogenic activities in the past half century, particularly in the increasing agricultural water consumption and conversion of wetlands to agricultural and other land uses.

Line141: the main research questions are not clear enough. My understanding is that you want to analyze how the combined wetland and reservoir operation can mitigate flood using modelling. So the (a) question should be written in that direction and the question (b) might be oriented towards the mitigation of wetlands and reservoir operations of future flood and drought.

Response: Thank you for highlighting the need to better describe the main research questions. We specified it in the introduction that:

We then applied it to a large river basin with abundant wetlands and a large reservoir, the Nenjiang River Basin in northeast China, to address a central question: Can the combining of wetlands with reservoir operation largely reduce the risk of future flood and droughts?

**Methods-**To many sections that are repetitive and could be deleted/incorporated into each other and with fewer sections.

Response: Thank you for indicating that this section needed improvement. We have largely adjusted the structure and revised text to make it more concise. Specifically in the following three aspects:

(1) Structurally, Section 2.2-2.3 of the original manuscript were reorganized into Section 2.3;

(2) Rewriting the text to detail how to couple wetlands and reservoir operation into hydrological modelling;

(3) Removal or merging of repetitive and irrelevant sentences/contents;

(4) Revise the text according to the constructive comments on Section 2.2-2.3.

Line 160: what is meant by "the wetlands and their contributing areas within the reaches"?

Response: We are sorry for unclear description of contributing drainage areas here. Since there is a detailed introduction to 'the wetlands and their contributing areas' later in the text, we have made a note here and a detailed introduction in subsequent text.

The sentence that contains the note is:

The wetlands and their contributing drainage areas (see Section 2.2.1 for specific definition) within the reaches monitored by the ten hydrological stations range from 14 to 23% and from 39 to 56% respectively, demonstrating the large wetland coverage of the NRB and its sub-basins (Table 1).

The detailed introduction in subsequent text: The contributing area of wetlands is defined as the sum of the area of all wetland RHHUs and upland RHHUs within their immediate catchment areas situated along active fill-spill pathways to the stream network (Evenson et al., 2016).

Line 164-165: you should delete that information. You already mention that the lower NRB is an important agricultural area.

Response: Deleted.

Line 166: what is meant by "ecological integrity"?

Response: We are sorry for unclear presentation here. We want to express the ecological integrity of wetlands. Based on your comments, it is modified as follows:

Therefore, understanding potential floods and hydrological droughts under future climate change is crucial for ensuring regional food security and wetland ecological integrity.

Line 168-171: these two sentences could be shortened into one.

Response: We thank your comment here and have condensed the two sentences into one as follows:

The area of wetlands in the NRB decreased by nearly 23% from 1978 to 2000 (Chen et al., 2021), with only 16.34% remaining today (Table 1), which largely degraded their services (Wu et al., 2021).

Line 177: Could you give a percentage of the total catchment instead? It would give more understanding of how important the area that drains into the reservoir if it was in percentage of the total catchment.

Response: We than your comment here and have detailed the drainage area of the reservoir as follows:

The drainage area of the reservoir accounts for 22.8% of the NRB.

Line 181: It would be easier to read the figure 1 if less background information was there. Also, the (b) and (c) part of the figure is missing.

Response: We apologize for the poor readability of Figure 1 and inconsistencies in the

representation here. We have redrawn Figure 1 and changed its caption as follows:

[Figure]

Figure 1. Location of the Nenjiang River Basin and the distribution of wetlands, river networks, Nierji Reservoir, and hydrological and meteorological stations within the basin.

Line 190: In relation to Table 1, could you use the ID of the hydrological stations in figure 1 as well? It could be good to have a link between figure 1 and table 1.

Response: We thank the reviewer for pointing this out. Corresponding to the ID of hydrological stations in Table 1, we redraw Figure 1. Please refer to our previous reply.

Line 193. Section 2.2 might be superfluous in the paper and not really relevant as the study approach should be clear from the beginning. You should consider integrating this information in coming sections.

Response: We agree that Section 2.2 (original manuscript) can be clear here and we have deleted irrelevant contents and integrated the key information into the Section 2.2 and 2.3.

Line 213: Section 2.3 is too short for the reader to understand how the hydrological modeling coupled the wetlands and reservoir operation.

Response: Thank you for highlighting the need to further detail how the hydrological modeling coupled the wetlands and reservoir operation. We agree with your constructive suggestion and have substantially modified this section, describing in detailed steps how to achieve the coupling of them. The revised contents are as follows:

We developed a spatially-explicit hydrological modeling framework that considers wetland hydrological processes and reservoir operations based on HYDROTEL model and reservoir simulation algorithms (Fig.2). Such a modeling framework was based on a distributed coupling implementation at watershed scale from upstream to downstream. Observed streamflow from seven hydrological stations (see hydrological stations 1-7 in Fig.1) located upstream of the Nierji Reservoir and three hydrological stations (see hydrological stations 8-10 in Fig.1) installed at downstream of the reservoir, respectively, were used to calibrate the HYDROTEL model. For the upstream Nierji Reservoir, we calibrated the HYDROTEL model against observed streamflow of seven hydrological stations with consideration of wetlands (i.e., hydrologic-wetlands model). Among the seven hydrological stations, the Nenjiang Station is located at the end of the upstream, where the simulated streamflow was taken as the inflow of the reservoir. We then computed the reservoir outflow using the simulated inflow, estimated lateral inflow and reservoir simulation algorithms (see Section 2.2.2), thereby integrating reservoir operation into the hydrologic-wetlands model to build a hydrologic-wetlands-reservoir model. Based on the calibrated hydrologic-wetlands-reservoir model, we simulated the outflow of the reservoir (Sect. 2.2.2), which was used as the input streamflow for downstream model calibration. For the downstream reservoir, we calibrated the hydrologic-wetlands-reservoir model against observed streamflow of Tongmeng, Fulaerji and Dalai Stations Based on this framework, the simulation of basin hydrological processes coupled with basin scale wetlands and reservoir operations were realized.

Line 220-221: please clarify "The simulated runoff simulated by hydrologic- wetland model at the reservoir outlet was replaced with the estimated reservoir outflow, thus integrating reservoir operation into the hydrological modeling (i.e. hydrologic-wetland-reservoir model)"

Response: We are sorry for unclear presentation here. We have revised this sentence to make it understandable as follows:

We then computed the reservoir outflow using the simulated inflow, estimated lateral inflow and reservoir simulation algorithms (see Section 2.3.2), thereby integrating reservoir operation into the hydrologic-wetland model to build a hydrologic-wetland-reservoir model.

Line 226: section 2.3.1 could be integrated to the overall section of 2.3.

Response: We agree with your suggestion and have integrated Section 2.3.1 into the overall Section 2.2.

Line 229-231: this sentences could be integrated and refereed to above sentence in line 227-229.

Response: We integrated the sentence into the above sentence as follows:

The PHYSITEL/HYDROTEL modeling platform coupled with two wetland modules (isolated and riparian wetlands) (Fossey et al., 2015a), which had been used to quantitatively evaluate the hydrological function of wetlands (Fossey and Rousseau, 2016, Fossey and Rousseau et al., 2016, Blanchette et al., 2019, Wu et al., 2020a; 2020b; 2021; 2022, Blanchette et al., 2022), was used to simulate hydrological processes, assess model performance and project future flood and drought conditions.

Line 243: Please clarify the concept of a "hydrologically equivalent wetland"?

Response: We thank your comments here and have clarified the concept of hydrologically equivalent wetland. Please refer to our previous reply.

Line 255 & 257: You quite some abbreviations in the text, could you delete some and use the full word instead? For instance RW and IW could just be fully written out in order to facilitate the reading.

Response: Thank you for your suggestion. We have reduced the number of abbreviations and only kept which full names are longer. Particularly, isolated and riparian wetlands and their contributing areas has revised be fully written out. Also, the full text has been adjusted accordingly.

Line 263: To better understand how you set the model, could you consider to more explicitly describe how you integrate the reservoir operation into one common section, ex: 2.3 together with the coupling of wetlands? Your section 2.3 is very long and could be more condensed into one section about the set up of the model.

Response: Thank you for highlighting the need to detail how to integrate the reservoir operation into hydrological model. We improved the current description in two aspects: (1) in section 2.2.2, we refined the simulation of the Nirvana reservoir operation; and (2) in section 2.2.3, we refined how the reservoir operation was coupled in the hydrological model.

Line 266: What are the three algorithms?

Response: We have added this point in the text. The three algorithms are as follows:

The first algorithm considers a case when we want to always release a constant amount over the simulation period. This constant amount is the target release that would cover all downstream demand for water, for instance for domestic use and/or irrigation. The second consider a case when we still want to release the target demand but we would also like to (1) apply some hedging (that is, an intentional reduction of the release - even if it would still be

feasible to release the target demand - aimed at saving more water and thus facing smaller deficits at later time); and (2) attenuate downstream peak flows for flood control purpose. The third algorithm, which was used in this study, dynamizes the operation rules.

Line 291-294: as the water level limit is always 216 m, could this sentence be rewritten in order to better understand the thresholds?

Response: Thank you for this suggestion. We have rewritten the sentence as follows:

The pre- and post-flood periods are June 1-20 and September 6-30, respectively, with a flood limited water level of 216.0 m; The main flood period is from June 21 to August 25, and the reasonable flood limited water level ranges from 213.4 m to 216.0 m and can be gradually increased.

Line 294-295: I sis not clear what the 25.3 % of the daily streamflow? Is it the average daily streamflow of the year or of the dry season?

Response: We are sorry for unclear presentation here. We consulted with the management of Nierji Reservoir and confirmed that it is the average streamflow during the dry season over the years. We have revised this sentence to make it clearly as follows:

During the dry season, the environmental flow was defined as 25.3% of the daily streamflow during the dry season over the years based on the designed operating curves of the reservoir operation chart.

Section 2.3.3 could be better integrated into the overall section of the model set up together with the other above sections.

Response: We agree with your suggestion and made several revisions in the following aspects: (1) Move 'description of driving datasets' to Section 2.1; (2) The rest of the content has been substantially revised to provide a clearer presentation of the model setup, calibration and validation process. The revised text is as follow:

2.2.3. Model calibration, validation and performance assessment

For all above scenarios, we calibrated the HYDROTEL model against observed streamflow at a daily time step over 8 years, including a 1-year warm-up (2010.10.01-2011.09.30) and a 7-year calibration (2011.10.01-2018.09.30) periods. The same model settings (i.e., key parameters, simulation periods, fitting algorithm, and objective function, etc.) were used for the calibration processes under the both presence and absence scenarios. Following Arsenault et al.(2018), the model was calibrated using full-time observations without additional validation, as the former allows for more reliable parameters and maximizes the accuracy of the model. The dynamically dimensioned search algorithm (DDS) developed by Tolson and Shoemaker(2007) was used to calibrate the 13 most sensitive parameters of the model as proposed by Foulon et al.(2018). Based on the maximizing of Kling-Gupta efficiency (KGE) (Gupta et al., 2009), automatic calibrations using DDS were carried out utilizing 10

optimization trials (250 sets of parameters per trial). Then, the best set of parameter values out the 10 trials were selected following Foulon et al.(2018). The KGE was chosen as the objective function because previous research has shown that it can improve flow variability estimates when compared to the NSE (Garcia et al., 2017; Fowler et al., 2018).

It should be noted that we calibrated the HYDROTEL model against observed streamflow under with and without wetland scenarios. For the without wetland scenarios are defined as follows: When the wetland modules are turned off in HYDROTEL, wetland areas are not removed, but they are treated as the land cover of saturated soils. Such a saturated soil is fixed and does not participate in hydrological processes such as water yielding and runoff routing, and thus their explicit storage properties are not accounted for in the modeling. This is a basic assumption that has been used in several studies using models such as SWAT (Liu et al., 2008; Wang et al., 2008; Evenson et al., 2015), Mike 11 (Ahmed, 2014) and HYDROTEL (Fossey et al., 2016; Fossey and Rousseau, 2016a, b; Wu et al., 2019, 2020a, 2021), to quantify the hydrologic services provided by wetlands (flood mitigation, flow regulation and baseflow support etc.).

To determine whether coupling the wetland module and the reservoir can improve the model performance, we compared (1) the efficiency of the model in simulating daily flow processes; and (2) the capability of the model to simulate floods and hydrological droughts in the presence or absence of the wetlands and the combination of wetlands and reservoir. Following the recommendations of N. Moriasi et al.(2007) and Moriasi et al.(2015), four performance criteria were selected to assess model performance with regards to simulated daily flows with and without the presence of the wetland modules and reservoir operation, namely the Nash-Sutcliffe efficiency (NSE) (Nash and Sutcliffe, 1970), Correlation Coefficient (CC), the root-mean square error (RMSE) and the percent bias (Pbias). We used multiple performance criteria because it may be unreliable to rely on a single objective function to determine whether the model performs well (Pool et al., 2018; Fowler et al., 2018; Seibert et al., 2018). It should be noted that although NSE as an objective function has shortcomings in model calibration, it can still provide an important reference for the evaluation of simulation results as a performance criterion as suggested by Moriasi et al. (2007, 2015). In addition, we compared model performance considering daily hydrograph changes. Furthermore, flood and drought features were extracted (see Sect. 2.4.2 and 2.4.3) and used to discern whether, and to what extent, the coupled wetland modules and reservoir simulations could improve the model's ability to simulate droughts and floods.

Section 2.4 and its subsection could also be better streamlined into one section describing the projection of future flood and drought.

Response: We agree with your constructive comments here and have streamlined Section 2.4 and its subsection into one section. Please refer to our previously revised text for details.

Line 383-389: Information that might be to detailed and should be mentioned rather swiftly than given too much attention.

Response: We acknowledge your suggestion to remove this section and move key information to other sections. In addition, we have moved the description of the datasets to Section 2.1 to make it easier for the reader to understand.

Line 433-434: repetitive sentence

Response: We thank your comments here and have rewritten the sentence as follows:

A threshold method was used to define hydrological drought events because it can determine the start and end of a hydrological drought event, which allows further assessment of drought characteristics, such as frequency, duration, and intensity of a drought event (Cammalleri et al., 2017).

Line 457-459: repetitive sentence of information already given before.

Response: Deleted.

**Results-**Your result should better reflect the modelling work. Instead of referring to the result figures in the supplementary materials, you should integrate them into the main text. In that way you should give more credit to how the model perform with couple wetland and reservoir operation that focusing on the projection of future flood and droughts.

Response: We would like to express our gratitude to you for the insightful comment. We have moved the key figures to the main text.

I would also recommend to use figure 5 &8 in text and leave figure 4 &7 in supplementary.

Response: We fully agree with your suggestion and have moved Figures 4 and 7 to the supplemental.

Figure 6 should be better explained and the explanation of the different plots (a-f) is missing.

Response: We thank your valuable comments here and have labeled the SSPs in Figure 5 (i.e., the Figure 6 in the original manuscript):

[Figure]

Figure 5. Historical and projected flood duration-peak flow-volume relationships at the Nenjiang (the first row) and Dalai (the second row) Station. The historical period refers to 1971-2020 and the near-future, mid-century and end-century refer to the 2026-2050, 2051-2075 and 2076-2100 under the Socioeconomic Pathways (SSP) 126 (the first column), SSP370 (the second column) and SSP585 (the third column) scenarios.

Line 619: please clarify the statement "droughts will equivalent to the historical period"
Here, it means that the number of droughts has no significant trend of increasing or decreasing.

Response: We are sorry for unclear state here. We intent to express that the number of droughts has no obvious increasing or decreasing trend. To better describe it, we revised the sentence as follows:

The number of droughts show no increasing or decreasing trend in the mid-century and end-century for the SSP126 scenario and in the mid-century for the SSP585 scenario compared to the historical period.

**Discussion**- Overall, your discussion section is less focused on the results of the work. You give statements that reflect the results but you tend to use that as reference to other research work. Sometimes, the discussion is too general and should be better connected to the aim and research questions and notably to the case study.

Response: Thank you for highlighting the need to improve discussion. This section has undergone extensive revision to better reflect our findings and tie them to the aim and research questions of this study. The specific changes are as follows.

(1) deleted section 4.5 from the original manuscript and moved the most relevant content to section 4.1; (2) substantially deleted section 4.2 from the original manuscript, retaining key

contents and moving them to the following sections; (3) carefully checked all the contents in the discussion and removed inappropriate case studies.

Line 714: you state that " such model performance improvement can minimize uncertainties" but is this the case for this analysis?

Response: We are sorry for unclear state here and have revised the sentence as follows:

Such model performance improvement can provide important information for developing downstream water resources management.

Line 736-744: It is not clear how this is related to the results of this work.

Response: We agree with your suggestion and have deleted these contents.

Section 4.5 could be deleted and integrated in the overall discussion of the results.

Response: Thank you for pointing this. We fully agree with you and have removed content that is not relevant to the purpose and results of the study, while keeping the most relevant content in other sections.

Line 831: "1 km resolution DEM" is information that should be mentioned earlier in the method part.

Response: We mentioned it in data description part (Section 2.1 Study area and datasets) as follows:

The land-use/land-cover types for 2015 (including wetland types), digital elevation models and digital elevation models with 1 km resolution were obtained from Resource and Environment Science and Data Center (https://www.resdc.cn/).

Technical corrections: typing errors, etc.

Line 219: delete "simulated"

Response: Deleted.

Line 432: delete "characteristics"

Response: Deleted.

Line 454: delete "The cumulative number of days during a drought event" as you repeat it just after the "i.e.," .

Response: Deleted.

Line 759: "needs an extensive assessment"

Response: Thanks for your kind reminder here. We have changed the sentence as follows:

Therefore, from perspective of NBS, it is important to further assess and understand their role

in improving basin resilience to water risks.

---

## Author Comment (AC4)

**Response to Anonymous Referee #1**

"Can the combining of wetlands with reservoir operation largely reduce the risk of future flood and droughts?" presents an original study aiming at discussing whether the combining of wetlands with reservoir operation can largely reduce the risk of future floods and droughts in the Nenjiang River Basin. The data of this paper is very comprehensive and the method is reasonable. The results are helpful for understanding the role of wetlands and reservoir operation in mitigating basin hydrological risks under climate change, which make this manuscript worth publishing. I recommend to accept this manuscript after minor revisions to address following general and specific comments.

> **Response**: We are grateful for Referee #1's very positive and constructive comments and suggestions. We have carefully revised this manuscript and provided the following point-to-point responses.

**General comments:**

Hydrological model is an important tool to understand wetland hydrological functions, and the same is true for observations. This cannot be neglected in research progress and discussion.

> Response: Many thanks for your comments here. We completely agree with your opinion here and have added relative contents in Introduction and Discussion as follows:
>
> Added contents in Introduction (Lines 68-76):
>
> To understand how and to what extent wetlands can mitigate hydrological processes, two approaches are commonly used: (i) description of individual wetland service at the field scale (e.g., Park et al., 2014) or wetlandscape scale (e.g., åhlén et al., 2022); (ii) assessment of wetland hydrological services at the regional/watershed scale (Fossey et al., 2016; Wu et al., 2020a, 2020b). However, the former approach only be achieved with field instruments and is mainly used to provide key parameters of wetland processes for model calibration (Fossey and Rousseau, 2016). Recently, several wetland hydrological models (e.g., SWAT model, PHYSITEL/HYDROTEL modeling platform) have been developed and applied to quantify hydrological function of wetlands, particularly the mitigation services on floods and droughts (Evenson et al., 2016; Evenson et al., 2018; Zeng et al., 2020; Fossey et al., 2015a).
>
> Added contents in Discussion (Lines 698-702):
>
> Therefore, there are ongoing efforts to obtain sufficient data on wetland area dynamics and evapotranspiration, water depth and volume, soil water content using multi-source remote sensing data and actual observations to better calibrate/validate watershed hydrological models, which are expected to better provide key parameters for further improving the model's capacity to capture flood and drought patterns.

I understand that wetland and reservoir can be regarded as green and gray infrastructure strategy respectively. The authors showed that the combination of them can experience the risks of hydrologic failure under future climate change. This is an interesting and important finding that can be further discussed beyond the current content.

> Response: We appreciate your comments here. Yes, wetland is one of green infrastructures and reservoirs are part of gray infrastructures. We found that based on existing reservoir operations

and wetland cover conditions, the combination of the two would experience a failure of flood and drought mitigation under future climate change. We agree with your point, that is, this is an interesting finding, and have enriched the discussion as follows (Lines 719-735):

Wetlands are typically viewed as green infrastructures and reservoirs are generally regarded as important gray infrastructures. Although our study showed that the combining of reservoirs and wetlands does not completely eliminate the risk of future hydrological extremes, they continue to play an important role that cannot be ignored. The reservoir's inherent constraints are one factor contributing to this likelihood of hydrological failure. This is because reservoirs only control floods and droughts that occur downstream of them, limiting their effects to the regional scale (Brunner, 2021). The regulation becomes less effective with distance increased due to "dilutions" effect caused by inflows from downstream tributaries (Guo et al., 2012). Reservoirs cannot, however, play a considerable role in basins where tributaries exist downstream, particularly those sub-basins that are vulnerable to drought and flooding. From these perspectives, widely distributed wetlands can provide a complementary and vital function by providing biological function and hydrological regulation in regions where reservoirs are unable to have an impact. On the other hand, the limited capacity of existing wetlands to regulate hydrology increases the risk of hydrological failure to some extent. This is because, compared with the historical period, the existing wetlands in the NRB have been seriously degraded, such as the weakening of the connectivity between riparian wetlands and the river channel, and the increased fragmentation of wetlands, among other changes (Chen et al., 2021). These degraded wetlands cannot play an effective role in mitigating floods and droughts under future climate change.

For different sub-periods under the constraints of three SSPs, the projected results are somewhat different, no matter floods and droughts, which can come into being some uncertainties and should be discussed.

Response: We appreciate your comments here. We fully agree with you that there will be some uncertainty in the predicted results based on different SSPs. However, we have described in the datasets section that the uncertainty can be reduced to a large extent by using the multi-model ensemble means approach. Moreover, this approach has been recognized and widely used by previous studies. Please see lines 193-196.

The assumption about without wetland scenarios, i.e., "wetland areas are not removed, but they are treated as the land cover of saturated soils". How the regulation function is not accounted for? I think there may be uncertainty here.

Response: We thank your comments here. Consistent with widely accepted conventional method, we evaluate the impact of wetlands on hydrologic processes using how the magnitude of storage capacity contributes to watershed water yielding and runoff routing. If the wetland area is not removed, and it is considered as saturated soils, this saturated soil is fixed and does not participate in hydrological processes such as water storage and flow yielding. Based on this, a simulation without considering the water storage function of the wetland is implemented. We appreciate your reminder here and have added a relevant description in the manuscript.

In Section 2.3.3, we specified this point as follows (Lines 340-343):

When the wetland modules are turned off in HYDROTEL, wetland areas are not removed, but they are treated as the land cover of saturated soils. Such a saturated soil is fixed and does not participate in hydrological processes such as water yielding and runoff routing, and thus their explicit storage properties are not accounted for in the modeling.

Minor errors and inadequacies in details that need to be double-check and revised. See specific comments below.

Response: We are grateful to your very helpful comments and suggestions, which have helped us clarify and improve the manuscript.

**Specific comments:**

Line 40 and 718. Diffenbaugh et al., 2015a and Diffenbaugh et al., 2015b are the same paper.

Response: Thanks for your reminder. We have revised it to the same paper.

Lines 42-43. Please cite the references in the main text correctly. Move "Güneralp et al., 2015" to the end of the sentence.

Response: Done.

Lines 45-46. Suggest to say global scale first and then regional scale.

Response: We thank your comment here. We have put the global scale before the regional scale.

Line 64. Inert "However" before unlike.

Response: Done.

Line 132. Delete "and storage".

Response: Done.

Line 137. Insert then after "We".

Response: Done.

Line 146. Insert hydrological after "future".

Response: Done.

Line 158-159. Three wetlands of international importance do not represent much. Please rewrite.

Response: We thank your comments here. We have carefully checked and rewrite the sentence as follows (Line 136-138):

The basin contains several important wetland conservation areas, among which Zhalong and Nanweng River Wetlands have been designated as a Ramsar Site of International Importance.

Line 160. What's the contributing drainage areas of wetlands?

Response: We are sorry for losing the definition of contributing drainage areas here. We have added relevant content and description to the main text as follows:

Lines 138-141. The wetlands and their contributing drainage areas (see Section 2.3.1 for specific definition) within the subbasins monitored by the ten hydrological stations range from

14 to 23% and from 39 to 56% respectively, demonstrating the large wetland coverage of the NRB and its sub-basins (Table 1).

Lines 247-249. The contributing area of wetlands is defined as the sum of the area of all wetland RHHUs and upland RHHUs within their immediate catchment areas situated along active fill-spill pathways to the stream network (Evenson et al., 2016).

Line 163-164. Songnun Plain or Songnen Plain? Please be consistent with Figure 1.

Response: It's the Songnen Plain. We have revised it.

Line 168. Revise "significantly" to largely.

Response: We have deleted this sentence to improve the readability.

Line 228-229. What are the two wetland modules. Please specify here.

Response: They are isolated wetlands and riparian wetlands. We have revised in the text as follows (Line 231-233):

The PHYSITEL/HYDROTEL modeling platform coupled with two wetland modules (isolated and riparian wetlands) (Fossey et al., 2015b), has been used to quantify hydrological function of wetlands (e.g., Fossey and Rousseau, 2016; Blanchette et al., 2019; Wu et al., 2023).

Line 231. The reference format is incorrect.

Response: Done.

Line 242. I still haven't seen a definition of contributing areas here.

Response: We have added relevant content, please see Lines 247-249.

Line 257. Kinematic wave equation, loss reference here.

Response: We have added the reference, please see Beven, 1981.

Line 274. What is the time-step length in this study?

Response: The time-step length is day. We have added it.

Lines 272-274, 277-279. Loss of units for formular parameters.

Response: We apologize for the lack of units here, which we have revised in the text.

Supporting references:

Åhlén, I., Thorslund, J., Hambäck, P., Destouni, G. and Jarsjö, J., 2022. Wetland position in the landscape: Impact on water storage and flood buffering. Ecohydrology, 15(7):e2458.

Beven, K.: Kinematic subsurface stormflow. 1981. Water Resour. Res. 17 (5), 1419-1424.

Brunner, M.I. 2021. Reservoir regulation affects droughts and floods at local and regional scales. Environ. Res. Lett. 16 (12), 124016.

Cook, B.I., Mankin, J.S., Marvel, K., Williams, A.P., Smerdon, J.E.Anchukaitis, K.J. 2020. Twenty-First Century Drought Projections in the CMIP6 Forcing Scenarios. Earth's Future 8 (6), e2019EF001461.

Guo, H., Hu, Q., Zhang, Q.Feng, S. 2012. Effects of the Three Gorges Dam on Yangtze River flow and river interaction with Poyang Lake, China: 2003-2008. J. Hydrol. 416, 19-27.

Karim, F., Kinsey-Henderson, A., Wallace, J., Arthington, A.H.Pearson, R.G. 2012. Modelling wetland connectivity during overbank flooding in a tropical floodplain in north Queensland, Australia. Hydrol. Process. 26 (18), 2710-2723.

Martel, J.L., Brissette, F., Troin, M., Arsenault, R., Chen, J., Su, T.Lucas Picher, P. 2022. CMIP5 and CMIP6 model projection comparison for hydrological impacts over North America. Geophys. Res. Lett. 49 (15), e2022GL098364.

Min, J., Paudel, R.Jawitz, J.W. 2010. Spatially distributed modeling of surface water flow dynamics in the Everglades ridge and slough landscape. J. Hydrol. 390 (1), 1-12.

Park, J., Botter, G., Jawitz, J.W.Rao, P.S.C. 2014. Stochastic modeling of hydrologic variability of geographically isolated wetlands: Effects of hydro-climatic forcing and wetland bathymetry. Adv. Water Resour. 69, 38-48.

Qing, Y., Wang, S., Zhang, B.Wang, Y. 2020. Ultra-high resolution regional climate projections for assessing changes in hydrological extremes and underlying uncertainties. Clim. Dyn., 1-21.

Zhang, B., Schwartz, F.W.Liu, G. 2009. Systematics in the size structure of prairie pothole lakes through drought and deluge. Water Resour. Res. 45 (4), 289, 2710-2723.

---

## Author Comment (AC5)

**Response to Anonymous Referee #2**

The authors developed a method for integrating wetlands and reservoirs into a semi-spatially explicit modeling framework (PHYSITEL/HYDROTEL) to project the magnitude, duration, and frequency of future floods and droughts in a northeast China river basin.

This is a straightforward paper that will be a great contribution toward our scientific understanding of how wetlands and reservoirs mediate droughts and floods, as part of the push for nature-based solutions. It also emphasizes the importance of integrating wetland and reservoir hydrological processes into watershed-scale models for large river basins.

> Response: We are grateful for Referee #2's thoughtful evaluation and support of our work. Our responses to all the comments are listed below in the order they appear.

Two main points for the authors to consider:

- The authors need to recheck the results text and compare that to their figures. The statements in the text often do not correspond to the figures – particularly those discussion future flood risks. Please see my specific comments below. Also, some figures (e.g., Figure 6) are not labeled, which makes it difficult to following along with some of the results.

  > Response: Thank you for your valuable comments and suggestions. We have carefully checked the full text and revised it in light of your specific comments below to ensure: (i) a consistency between the statements of the text and the figures; (2) All figures are clearly marked.

- The paper is generally well-written, but please re-review it for grammatical errors. I made a few specific suggestions below, but there are several others throughout.

  > Response: We thank your comment here. We regret that there were grammatical errors. The paper has been carefully revised by all authors to improve the grammar and readability.

I also have some general suggestions, including:

Line 43 – Delete "In the future" to make verb tenses correct.

> Response: Changed as suggested.

Line 46 – Move "disaster-related" to before the word "loss".

> Response: Changed as suggested.

Line 107 – Change to "included", not "including".

> Response: Changed as suggested.

Lines 107-112 – Not 100% following this statement: As I read it, it states that integrating wetland and reservoir hydrological processes in the calibration process increases model error and uncertainties but that integrating wetlands and reservoirs (without processes?) minimizes uncertainties and improved model performance? The studies cited *do* integrate some hydrological processes of wetlands and reservoirs, and overall these statements do not seem to align. Could you please clarify this statement?

Response: We are sorry for unclear state here. What we want to state in the text is that: disregarding of the wetlands or reservoir operation would add significant error and larger uncertainties to simulate hydrologic processes; while integrating the wetlands or reservoir operation alone into watershed-scale hydrologic models may largely minimize uncertainties and improve model performance. We have rephase the sentences as follows (Lines 112-121):

Recent studies have suggested that disregarding of the wetlands or reservoir operation would add significant error and larger uncertainties to simulate hydrologic processes (Ward et al., 2020; Brunner et al., 2021). Because wetlands are often abundant across many landscapes, making their water storage and cycling fundamental to estimating a watershed's water balance (Rains et al., 2016; Lee et al., 2018). Therefore, missing this component of water balances could potentially lead to disproportionately large model errors (Rajib et al., 2020). Consequently, integrating the wetlands (Rajib et al., 2020; Golden et al., 2021; Fossey et al., 2015a) or reservoir operation (Zhao et al., 2016; Dang et al., 2020; Yassin et al., 2019) alone into watershed-scale hydrologic models may largely minimize uncertainties and improve model performance.

Figure 1. The caption says the figure shows elevation, isolated wetlands, riparian wetlands, their drainage areas, and land-use types. Out of these listed, I only see lumped "wetlands" and nothing else in the figure legend indicating the other listed characteristics of the watershed. Please amend the figure or the caption.

Response: We apologize for the inconsistencies in the representation here. We have changed the caption as follows:

Figure 1. Location of the Nenjiang River Basin and the distribution of wetlands, river networks, Nierji Reservoir, and hydrological and meteorological stations within the basin.

Lines 240-241: Are only the pixels adjacent to the hydrographic network considered riparian and all others are isolated? Please include that information here. Also, in the subsequent lines, you may want to define for the reader what the HEW concept is and to specifically mention the "lumped" nature of HEWs. Just a few extra words are needed here for clarity.

Response: We thank the reviewer for pointing this out. We have rephase this sentences and added additional statement to clear this as follows (Lines 241-245):

In addition, the PHYSITEL platform distinguishes wetlands from other land-use types, and then classifies both isolated and riparian wetlands based on an adjacency threshold (i.e., percentage of pixels in contact) between the wetlands and the river network (Fossey et al., 2015). Specifically, if more than the adjacency threshold (e.g., 1%) of wetland pixels are connected to the river network, they are considered as pixels of a riparian wetlands; otherwise, they are referred to as isolated wetlands.

Lines 256-261: Within the RHHU, isolated wetlands cannot hydrologically connect to RWs, correct, because of spatial lumping? May be worth mentioning here.

Response: Thanks for your comments. Indeed, isolated wetlands cannot hydrologically connect to RWs within a RHHU. This is because we integrated RWs and IWs based on the concept of hydrologic equivalent wetland (HEW) developed by Wang et al. (2008). We have added additional statement to clear this point as follows (Lines 2674-278):

It should be mentioned that the HEW concept developed by Wang et al (2008) served as the foundation for the integration of RWs and IWs into the modeling framework. This concept contends that the features of one HEW (also known as an isolated wetland or riparian wetland) are equivalent to the sum of the characteristics of each wetland inside a RHHU (which could either be hill slopes or elementary sub-watersheds related to one river segment). The following premises apply to this concept: (i) only one isolated and/or riparian HEW per RHHU; (ii) one HEW can be fully integrated within a RHHU; (iii) isolated HEW parameters must be numerically integrated; and (iv) riparian HEW parameters must be numerically integrated and spatially integrated (i.e., located in a specific location on the river segment). Therefore, IWs and RWs do not appear to have direct hydrological connection within a RHHU. However, IWs also have hydrological interactions with RWs through vertical water balance processes and fill-spill processes (Fossey et al., 2015).

Lines 301-302: What did you do with the data once overlaid? Did the 2015 wetland distribution maps trump the land-use/land-cover data (meaning did you use that instead)? Please mention here how the wetlands were represented once the overlay with the lu-lc data happened.

Response: We thank your kind reminder here. We apologize for the unclear presentation here. The land-use/land cover types data we collected contains wetland types and can be directly used as input data for PHYSITEL without additional overlay. We have deleted the sentence and revisited another sentence as follows (Lines 166-169):

The land-use/land-cover types for 2015 (including wetland types), soil texture, digital elevation models, digital elevation models and drainage network were obtained from Resource and Environment Science and Data Center (https://www.resdc.cn/).

Lines 320-322: So to be clear, you have two calibrated models: one with wetlands and one without? I read later (lines 367-369) that the wetland- and reservoir-integrated model is used for future flow projections. I would mention that here, too, since it's not clear here why there were two model calibrations.

Response: We thank your valuable comments here. This is not a simulation with and without wetland scenarios, but with and without wetland-reservoir scenarios. We apologize for the unclear presentation here. We have revised the sentence as follows (Lines 219-221):

For the downstream reservoir, we calibrated the hydrologic-wetlands-reservoir model against observed streamflow of Tongmeng, Fulaerji and Dalai Stations.

Line 340: Had you considered using a behavioral parameter set so that output uncertainty bounds could be produced?

Response: We thank your comments here. We considered behavioral parameter set when conducting model calibration. Because a discussion on this point is beyond the scope of this paper, we recommend you to read the work of Foulon and Rousseau, (2018). However, we thank your kind reminder here and added relative reference to detail this point.

Then, the best set of parameter values out the 10 trials were selected following Foulon et al. (2018).

Line 349: Why use the NSE here when in lines 340-342 you argued against it? Please add the

rational here.

Response: We thank the reviewer for pointing this out. Firstly, compared with NSE, using KGE as the objective function can improve the calibration efficiency of hydrological model. Specifically, KGE combines the three components of Nash-Sutcliffe efficiency (NSE) of model errors (i.e., correlation, bias, ratio of variances or coefficients of variation) in a more balanced way, has been widely used for calibration and evaluation hydrological models in recent years (Liu, 2020). Therefore, we against NSE in lines 340-342. Second, we used multiple performance criteria (including NSE, CC, RMSE and Pbias) because it may unreliable to rely on a single objective function to determine whether the model performs well (Pool et al., 2018; Fowler et al., 2018; Seibert et al., 2018). Here, the NSE is mainly used to evaluate the simulation efficiency of the calibrated model. Based on these, we have changed "objective function" to "performance criteria", to distinguish the utility of NSE in the text. Then, we have added a sentence to state the rational here as follows (Lines 359-361):

It should be noted that although NSE as an objective function has limitation in model calibration, it can still provide an important reference for the evaluation of simulation results as a performance criterion as suggested by Moriasi et al. (2007, 2015).

Line 351: Add "be" in front of unreliable.

Response: Changed as suggested.

Line 498: Replace "Specially" with "In fact".

Response: Changed as suggested.

Line 565: Add "and" in front of "increase"

Response: Changed as suggested.

Paragraph starting on 559 – It would be helpful to point to the exact figure (e.g., Figure 4b or Figure 5d) when describing results so that the reader can follow along closely with the text. It will also help to correct some of the errors listed below.

Response: We agree with your constructive suggestion and have detailed all figures referenced in the text.

Lines 566-568: The near-future flood volumes appear to remain the same for the near future SP126 and SP370 pathways in Figure 4g and do not decrease in Figure 5c = the statement and figures do not seem to correspond. Also, what does varying contrarily mean? Check these statements and please clarify. Similarly...

Response: We apologize for unclear statement here. We have rewritten this part as follows (Lines 534-539):

Flood volume shows divergent change trend under the three SSPs. For the SSP126 scenario, flood volume will grow in the near-future and diminish in the mid- and end-century. Flood volume will decrease in the near-future, increase in the mid-century, and increase slightly in the end-century under the SSP370 scenario. However, following an apparent reduction in the near-future, flood volume is anticipated to have no discernible change trend in the mid-century and a clear increasing trend in the end-century for the SSP585 scenarios.

Lines 568-569: Flashiness does *not* appear to follow the trends stated in the text, compared to Figure 5d. Flashiness also doesn't increase substantially for SSP370, as stated in the text, in the near future. It decreases in Figure 5d. Also, near-century needs to be changed to near future on Line 569.

Response: We are sorry that the representation does not correspond to the Figures. Here, lines 568-569 correspond to Figure 5h rather than Figure 5d. For your convenience in reading and understanding, we have specified all the figures in the text in this paragraph.

Check sentence on line 571-572 – "...flashiness will experience a considerable increase of flashiness..."???

Response: We apologize for the typographical error here. We have deleted "of flashiness" in this sentence.

Figure 6: The SSPs are not labeled (I *think* the columns represent SSPs?), so it's difficult to interpret Figure 6. Please re-do the figure with labels and check/re-write, if necessary, the paragraph from 584-596 and the one from 603-612 to correspond with Figure 6.

Response: We thank your valuable comments here. The SSPs has been labeled in Figure 6. Further, we have carefully checked the text and revised few sentences to keep correspondence.

The labeled Fig.6 is as follows:

[Figure]

Lined 615: Add "d" to increase.

Response: Changed as suggested.

Line 619: Delete "y" from clearly and add "be" in front of equivalent.

Response: Changed as suggested.

Line 625: Change "shorting" to "shortening".

Response: Changed as suggested.

Line 627: capitalize SSP in ssp126.

Response: Changed as suggested.

Figure 9: SSPs need to be labeled on the figure itself.

Response: Changed as suggested.

Lines 627-629: Definitely true, though that is unclear in the violin plots (Figure 7). I would suggest considering moving Figure 7 and Figure 4 to the supplemental. They don't really add much and can partially confuse the story because the results are visually clear at that resolution.

Response: We fully agree with your suggestion and have moved Figures 4 and 7 to the supplemental.

Lines 847-878: It seems the study explored how the integration of wetlands and reservoirs affect the streamflow test statistics for a river basin modeling framework and how climate-change induced floods and droughts can be projected using this wetland- and reservoir- integrated model. That's slightly more nuanced than what is currently stated and seems a bit more correct?

Response: We apologize for the logical error in the statement here. This study explored "How will estimated future flood and drought risks be changed by considering the combined effects of wetlands and reservoirs?". To achieve this objective, we first developed a framework of hydrological modeling coupled with wetland modules and reservoir operation scenarios, and then applied it to the Nenjiang River Basin. We have rewritten the first two sentences in Conclusion, as follows (Lines 772-775):

This study projected future flood and drought risks by considering the combined impacts of wetlands and reservoirs. To achieve this, we developed a hydrological modeling framework coupled wetlands and reservoir operations and then applied it in a case study involving a 297,000-km2 large river basin in northeast China.

**References:**

Dang, T.D., Chowdhury, A.F.M.K. and Galelli, S., 2020. On the representation of water reservoir storage and operations in large-scale hydrological models: implications on model parameterization and climate change impact assessments. Hydrology and Earth System Sciences, 24(1): 397-416.

Foulon, É. and Rousseau, A.N., 2018. Equifinality and automatic calibration: What is the impact of hypothesizing an optimal parameter set on modelled hydrological processes? Canadian Water Resources Journal / Revue canadienne des ressources hydriques, 43(1): 47-67.

Fossey, M., Rousseau, A.N., Bensalma, F., Savary, S. and Royer, A., 2015. Integrating isolated and riparian wetland modules in the PHYSITEL/HYDROTEL modelling platform: model performance and diagnosis. Hydrological Processes, 29(22): 4683-4702.

Golden, H.E., Lane, C.R., Rajib, A. and Wu, Q., 2021. Improving global flood and drought predictions: integrating non-floodplain wetlands into watershed hydrologic models.

Environmental Research Letters, 16(9): 091002.

Liu, D., 2020. A rational performance criterion for hydrological model. Journal of Hydrology, 590: 125488.

Rajib, A., Golden, H.E., Lane, C.R. and Wu, Q., 2020. Surface Depression and Wetland Water Storage Improves Major River Basin Hydrologic Predictions. Water Resources Research, 56(7): e2019WR026561.

Wang, X., 2008. Using Hydrologic Equivalent Wetland Concept Within SWAT to Estimate Streamflow in Watersheds with Numerous Wetlands. Transactions of the ASABE, 51(1): 55-72.

Ward, P.J. et al., 2020. The need to integrate flood and drought disaster risk reduction strategies. Water Security, 11: 100070.

Yassin, F. et al., 2019. Representation and improved parameterization of reservoir operation in hydrological and land-surface models. Hydrology and Earth System Sciences, 23(9): 3735-3764.

---

## Referee Report (RR1)

Review of "Can the combining of wetlands with reservoir operation largely reduce the risk of future flood and droughts?"

Wu et al. deals with flood and hydrological drought risk under future climate change by considering the combined effects of wetlands and reservoirs. The author develops a novelty framework coupling wetlands and reservoir operations with a semi-spatially explicit hydrological model and then apply it in a case study basin to project future flood and drought characteristics under CMIP6 scenarios. This topic will be of general interest in view of new insights into the role of grey-green infrastructure such as wetlands-reservoir in mitigating hydrological extremes. The manuscript is well written, the figures are excellent, and the contribution is clear presented. Overall, I think the topic falls into the scope of HESS.

I have a concern with the role of tributary reservoirs in flood and drought management, and some suggestions for improving the manuscript (see details below). Overall, I recommend minor revisions.

Major comments
In this study, the Nierji reservoirs located in mainstream were mainly considered, but the water regulation role of tributary reservoirs cannot be ignored. Some sub-basins of the Nengjiang River basin also have reservoirs and can make a certain degree of impact on streamflow (Meng et al., 2019), i.e. on the characteristics of floods and droughts. Would the risk of future floods and droughts be different if coupled simulations of multiple reservoirs-wetlands in the mainstream and tributaries were carried out? Such point is important for the next work of multi-objective optimization algorithm. Although the authors did not state such concern may be due to data limitations, I think this point is crucial to mention and discuss.

The figure has problems such as title and content not corresponding, and wrong unit labeling, which need to be carefully checked and revised to improve readability.

Specific comments
Line 180-181. Latitude information is missing in Fig.1. The title and content of Figure 1 do not match. Where are the subfigures (a), (b) and (c)? Please rewrite the title.
Line 207. Insert 'basin' between 'a' and 'hydrological' to maintain consistency with subgraph (a).
Lines 297-298. What are the resolution of the raster datasets used?
Line 309. Delete the sentence 'Of the ten stations, seven are located upstream of the Nierji Reservoir.' to avoid repetition with the first sentence of the next paragraph.
Line 342. This is the first occurrence of the NSE and needs to be stated in full name, not in the second occurrence of the NSE in line 349.
Line 369. What is SSP, please specify the full name when it first appears.
Line 394.Change 'used' to 'adopted'.
Line 411. What is the specific time period of the historical period?
Line 595. Insert 'scenarios' at the end of this sentence.
Line 597. The Y-axis is 'Peak flow' not the 'Peakflow'; and should be consistent with the other figures and main text.
For Fig.6, Fig.8, Fig.9, and Fig.A3, the unit of drought deficit should be 'm$^3$' rather than 'm$^3$/s'

based on equation (4).

Lines 903 and 908. Change 'wetlands/wetlands and reservoir' to 'wetlands/wetlands-reservoir'

Supporting references:

Meng, B., Liu, J. L., Bao, K., & Sun, B. (2019). Water fluxes of Nenjiang River Basin with ecological network analysis: Conflict and coordination between agricultural development and wetland restoration. Journal of Cleaner Production, 213, 933-943.

---

## Author Response (AR2)

**Response to Anonymous Referee**

**Response to Anonymous Referee #1**

Review of "Can the combining of wetlands with reservoir operation largely reduce the risk of future flood and droughts?"

Wu et al. deals with flood and hydrological drought risk under future climate change by considering the combined effects of wetlands and reservoirs. The author develops a novelty framework coupling wetlands and reservoir operations with a semi-spatially explicit hydrological model and then apply it in a case study basin to project future flood and drought characteristics under CMIP6 scenarios. This topic will be of general interest in view of new insights into the role of grey-green infrastructure such as wetlands-reservoir in mitigating hydrological extremes. The manuscript is well written, the figures are excellent, and the contribution is clear presented. Overall, I think the topic falls into the scope of HESS.

I have a concern with the role of tributary reservoirs in flood and drought management, and some suggestions for improving the manuscript (see details below). Overall, I recommend minor revisions.

> **Response**: We are grateful for Referee #1's very positive and constructive comments and suggestions. We have carefully revised this manuscript and provided the following point-to-point responses (highlighted in yellow).

**Major comments**

In this study, the Nierji reservoirs located in mainstream were mainly considered, but the water regulation role of tributary reservoirs cannot be ignored. Some sub-basins of the Nengjiang River basin also have reservoirs and can make a certain degree of impact on streamflow (Meng et al., 2019), i.e. on the characteristics of floods and droughts. Would the risk of future floods and droughts be different if coupled simulations of multiple reservoirs-wetlands in the mainstream and tributaries were carried out? Such point is important for the next work of multi-objective optimization algorithm. Although the authors did not state such concern may be due to data limitations, I think this point is crucial to mention and discuss.

Response: Many thanks for your constructive comments here. We completely agree with your opinion here and have added relative contents in Discussion to clear state this point here as follows (Lines 711-719):

Furthermore, several reservoirs and a large number of wetlands are spread throughout the NRB's tributaries (Meng et al., 2019), which individually and together play an essential role in drought and flood risk reduction. We only investigated the impacts of mainstream reservoirs and wetlands on drought and flood risk due to a lack of sub-basin reservoir operation observations. As a result, future integrated wetland-reservoir simulations of all mainstream and tributaries for flood-drought risk assessment will be done based on further data collection. Since the Nierji reservoir is the largest in the NRB and has the most influence on the mainstream runoff regime, our findings based on the simulation of Nierji reservoir and wetlands can give new insights into future floods and droughts, as well as provide important support for future hydrological extremes adaptation.

Hydrological model is an important tool to understand wetland hydrological functions, and the same is true for observations. This cannot be neglected in research progress and discussion.

Response: We thank your comments here. We fully agree with you that both hydrological and observation are important tools that can be used for understanding wetland hydrological functions. We have added relevant content in research progress and discussion to state this point.

Lines 72-74:

However, the former approach can only be achieved with field observation with instruments and is mainly used to provide key parameters of wetland processes for model calibration (Fossey and Rousseau, 2016).

Lines 696-702:

In addition, due to a lack of wetlands water balance monitoring data, this study only used river station data (which only considered the cumulative hydrologic effect of upstream wetlands) for model calibration. Therefore, there are ongoing efforts to obtain sufficient observations on wetland area dynamics and evapotranspiration, water depth and volume, soil water content and actual observations to better calibrate/validate watershed hydrological models, which are

expected to better provide key parameters for further improving the model's capacity to capture flood and drought patterns and better serve basin water management.

The figure has problems such as title and content not corresponding, and wrong unit labeling, which need to be carefully checked and revised to improve readability.

Response: We thank your comments here. We have carefully checked all the figures to make sure the titles and contents are consistent, and we changed the wrong unit labeling.

Lines 158-160: Fig.1.

Line 207. Insert 'basin' between 'a' and 'hydrological' to maintain consistency with subgraph (a).

Response: Done.

Lines 297-298. What are the resolution of the raster datasets used?

Response: We thank your comments here. The resolution of the raster datasets is 1km. We have added the content as follows s (Lines 168):

The land-use/land-cover types for 2015 (including wetland types), digital elevation models and digital elevation models with 1 km resolution were obtained from Resource and Environment Science and Data Center (https://www.resdc.cn/).

Line 309. Delete the sentence 'Of the ten stations, seven are located upstream of the Nierji Reservoir.' to avoid repetition with the first sentence of the next paragraph.

Response: Done.

Line 342. This is the first occurrence of the NSE and needs to be stated in full name, not in the second occurrence of the NSE in line 349.

Response: We thank your valuable comments here and have added the full name in the sentence as follows s (Lines 337):

The KGE was chosen as the objective function because previous research has shown that it can improve flow variability estimates when compared to the Nash-Sutcliffe efficiency (NSE) (Fowler et al., 2018; Garcia et al., 2017).

Line 369. What is SSP, please specify the full name when it first appears.

Response: We thank your comments here and have added the full name in the sentence as follows (Lines 178-179):

In this study, we drove hydrological model using five GCM projections (GFDL-ESM4, IPSL-CM6A-LR, MPI-ESM1-2-HR, MRI-ESM2-0, UKESM1-0-LL) under three Socioeconomic Pathways (SSPs)

Line 394. Change 'used' to 'adopted'.

Response: Done.

Line 411. What is the specific time period of the historical period?

Response: We appreciate your comments here and have added the historical period in the when it first appear in the manuscript follows (Lines 178-179):

The flood and drought characteristics were then compared against historical periods (1971-2020) to discern how future hydrological extremes will be changed under the influence of wetlands and reservoirs (see Part II in Fig.2).

Line 595. Insert 'scenarios' at the end of this sentence.

Response: Done.

Line 597. The Y-axis is 'Peak flow' not the 'Peakflow'; and should be consistent with the other figures and main text.

Response: We thank your comments here and have modified Figure 7 to keep the figure and text consistent.

[Figure]

Figure 7. Historical and projected flood duration-peak flow-volume relationships at the Nenjiang (the first row) and Dalai (the second row) Station. The historical period refers to 1971-2020 and the near-future, mid-century and end-century refer to the 2026-2050, 2051-2075 and 2076-2100 under the Socioeconomic Pathways (SSP) 126 (the first column), SSP370 (the second column) and SSP585 (the third column) scenarios.

For Fig.6, Fig.8, Fig.9, and Fig.A3, the unit of drought deficit should be 'm3' rather than 'm3/s' based on equation (4).

Response: We thank your comments here and have modified Fig.6, Fig.8, Fig.9, and Fig.A3 with the right unite. The revised figures are as follows:

[Figure]

Figure 7. Historical and projected flood duration-peak flow-volume relationships at the Nenjiang (the first row) and Dalai (the second row) Station. The historical period refers to 1971-2020 and the near-future, mid-century and end-century refer to the 2026-2050, 2051-2075 and 2076-2100 under the Socioeconomic Pathways (SSP) 126 (the first column), SSP370 (the second column) and SSP585 (the third column) scenarios.

[Figure]

Figure 9. Historical and projected duration-deficit relationship of each hydrological droughts

at the Nenjiang (the first row) and Dalai (the second row) Station. The historical period refers to 1971-2020 and the near-future, mid-century and end-century refer to the 2026-2050, 2051-2075 and 2076-2100 under the Socioeconomic Pathways (SSP) 126 (the first column), SSP370 (the second column) and SSP585 (the third column). The dark yellow lines in the horizontal and vertical directions refer the 95% threshold lines for drought deficit and duration values, respectively. I, II, III and IV refer to short-term light droughts, long-term light droughts, short-term severe droughts, and long-term severe droughts, respectively.

[Figure]

Figure. A4. Historical and projected hydrological drought characteristics (the number of droughts, annual drought days, duration, and deficit) at the Nenjiang (the left column) and Dalai (the right column) Station. The historical period refers to 1971-2020 and the near-future, mid-century and end-century refer to the 2026-2050, 2051-2075 and 2076-2100 under the Socioeconomic Pathways (SSP) 126, SSP370 and SSP585 scenarios. Note that the wider the violin plot, the higher the density.

Lines 903 and 908. Change 'wetlands/wetlands and reservoir' to 'wetlands/wetlands-reservoir'

Response: Done.

**Response to Anonymous Referee #2**

This is an important manuscript adding to the literature focused on understanding the role of wetlands and reservoirs in flood-risk prevention. The paper is much improved from the original version, particularly in the methodological descriptions (which are now very clear), scientifically sound, and the authors largely addressed my original comments. I have a few very minor comments to consider prior to publication.

Response: We are thankful to Referee 2 for the time and constructive comments and suggestions on our manuscript. Please find our detailed replies to minor comments below (highlighted in Gray).

Title – remove "largely" because it is not very descriptive

Response: Done.

Line 69 – add "s" onto services

Response: Done.

Line 72 – add "can" in front of "only"

Response: Done.

Lines 97-99 – not quite a complete sentence

Response: Done.

Because wetlands are often abundant across many landscapes, which make their water storage and cycling fundamental to estimate a watershed's water balance (Rains et al., 2016; Lee et al., 2018).

Line 105 – delete "a"

Response: Done.

Quick suggestion: Review the paper one more time for small grammatical errors. It would be well

worth it before returning it for the second revision. From here forward, I'll stop suggesting these edits directly.

Response: We thank your comments here and have carefully checked the full text to correct the small grammatical errors.

Lines 284-286 – repetition in sentence

Response: We thank your valuable comments here and have revised the sentence as follows.

Based on the simulated runoff at the inlet (the Nenjiang Station), lateral inflow, and the schemes of reservoir operation, we estimated the reservoir outflow using the ResSimOpt-Matlab software package developed by Dobson et al., 2019.

Line 585 – The sentence, "However, the fact that extreme flood events that will still occur in the future" should be changed to "However, extreme floods will still occur in the future".

Response: Done.